# Maternal thyroid hormone is required to develop the hindbrain vasculature in zebrafish
Marlene Trindade[1,2], Nádia Silva [1], Joana Rodrigues [1], Koichi Kawakami [3] & Marco A. Campinho [1,4] ✉

Thyroid hormone (TH) signaling is important and necessary for proper neurodevelopment. Inadequate levels of maternally derived THs (MTH) supply affect target gene expression profiles, which are fundamental for the brain's normal growth, maturation, and function. The monocarboxylate transporter 8 (SLC16A2, MCT8) is the main TH transporter present in the brain during embryonic development, and mutations in this transporter lead to a rare and debilitating human condition known as the Allan-Herndon-Dudley Syndrome (AHDS). This mutation affects the capacity for intracellular transport of the hormone, leading to impaired brain development that constitutes the main pathophysiological basis of AHDS. Like humans, zebrafish embryos express *slc16a2* that transports exclusively T3 at zebrafish physiological temperature. Studies in zebrafish Mct8 knockdown (KD) models found impaired hindbrain vasculature development. Here, using zebrafish Mct8 KD and knockout (KO) models, we shed light on the maternal T3 (MT3)-dependent developmental mechanism behind hindbrain vasculature development. We first demonstrate that MT3-regulates hindbrain *vegfaa* expression. We provide evidence that hindbrain neurons are not the source of *vegfaa*, instead, restricted *pax6a*+ neuroprogenitor cells (NPCs) instruct central arteries (CtAs) ingression into the hindbrain. Therefore, MT3 acts as an integrator, providing the regulatory cues necessary for the timely ingression of the CtAs into the hindbrain.

Thyroid hormones (THs) are essential for human central nervous system (CNS) development[1]. In humans, an inadequate supply of maternally derived THs (MTHs) during prenatal stages causes several neurological impairments, affecting a newborn's psychomotor and cognitive development. The most severe condition results from mutations of the *monocarboxylate transporter 8* (*MCT8*) gene, leading to a rare X-linked neurodevelopmental disorder, the Allan-Herndon-Dudley Syndrome (AHDS)[2,3]. AHDS patients present different severities of the condition that have been correlated with the particular mutations they carry[4–8]. One of the most prevalent pathological characteristics of AHDS is the low myelination levels found by brain magnetic resonance imaging (MRI) of these patients[8–10].

The zebrafish *slc16a2* (*mct8*) gene is also expressed during embryogenesis[11,12]. Notably, at physiological zebrafish temperature, Mct8 can only transport 3,5,3'-triiodothyronine (T3), showing that during zebrafish embryogenesis, the only form of MTH transported is T3[11]. New

evidence in human brain organoids derived from AHDS patient's cells also shows impaired maternal T3 (MT3) transport by MCT8, highlighting that the lack of T3 is responsible for the pathophysiological outcome behind AHDS[13]. The emergence of zebrafish as a model species for AHDS brought into light another pathological aspect not previously found or looked for in humans or murine models. In these zebrafish models[12,14], pronounced changes were found in the development of the hindbrain vasculature, which gave rise to the blood-hindbrain barrier (BHB). However, no TH signaling genes, including *mct8*, were expressed in zebrafish endothelial cells during development[12], arguing that these cells are not direct targets of T3. In AHDS patient-derived induced pluripotent stem cells (iPSCs), the endothelial blood-brain barrier (BBB) in vitro differentiation suggests that it is not dependent on *MCT8* expression and T3 function[15], further indicating that endothelial BBB development is independent of T3 signaling. Although this study has shown that endothelial differentiation is not dependent on T3, it does not answer whether T3 is necessary for adequate brain vasculature

[1]Algarve Biomedical Center-Research Institute, Universidade do Algarve, Faro, Portugal. [2]Center for Marine Sciences of the University of the Algarve, Faro, Portugal. [3]Laboratory of Molecular and Developmental Biology, National Institute of Genetics, Mishima, Shizuoka, Japan. [4]Faculty of Medicine and Biomedical Sciences, University of Algarve, Faro, Portugal. ✉e-mail: macampinho@ualg.pt

development. Nonetheless, new evidence has emerged that shows impaired cortex vascularization in an 11-year-old patient with AHDS and also in a murine model of AHDS (*mct8/dio2* knockout mice)[16].

Previous evidence has pointed out that THs' effect on angiogenesis is mediated by the αvβ3 integrin that serves as a membrane receptor for T4[17]. TH has been found to promote angiogenesis[18] via the PDGF-Akt pathway in adult mice hearts. In squamous cell carcinoma, TH promotes tumor angiogenesis[19]. In mice, in vivo and in vitro evidence shows that TH activating enzyme, *Deiodinase type 2*, is essential for muscle vascularization[20]. In all these cases, a common aspect of TH-induced angiogenesis involves downstream up-regulation of *vascular endothelial growth factor-A* (*VEGFA*) expression and secretion[18–20]. Post-natal induced hypothyroid mice present decreased brain vascularization of the CNS. Both in vivo and in vitro evidence shows that TH action is mediated by TH receptors and leads to increased *VEGFA* expression[21]. Together, these studies suggest that TH is essential for vascularizing several organs via *VEGFA* regulation.

Here, we provide evidence that MT3 is essential for the complete vascularization of the zebrafish hindbrain during embryogenesis. This action of MT3 is mediated via *mct8* in a discrete *pax6a*+ neuroprogenitor cell (NPC) population, from 30 to 48 h post-fertilization (hpf), which are the source of *vascular endothelial growth factor aa* (*vegfaa*) that instruct the ingression of the sprouting central arteries (CtAs) into the hindbrain. Moreover, our findings suggest that the timing and combination of the expression of the *thyroid hormone receptor aa* (*thraa*) and *thyroid hormone receptor ab* (*thrab*) in *pax6a* + NPCs are essential for the chronological ingression of the sprouting CtAs into the hindbrain. Our findings show an unexpected role for MT3 in CNS angiogenesis in the zebrafish AHDS model. This work sheds light on the genetic and cellular mechanisms driven by MT3 in developing the brain vasculature.

## Materials and methods
### Zebrafish husbandry and spawning
Adult wildtype (AB strain), mutant and transgenic zebrafish were maintained in standard conditions in the Center of Marine Sciences (CCMAR) fish facility at the University of Algarve (Portugal) as described[12]. The zebrafish lines used were: wild-type AB strains, *Tg(fli1:EGFP)*[22], *Tg(kdrl:CaaX-mCherry)*[23], *Tg(pax8:DsRed)*[24], *Tg(gSA2AzGFF306A)* (illuminating Cpne4+ cells)[25,26] and *mct8* knockout (KO, *mct8*[(-/-)])[27]. All experiments and fish husbandry were carried out by fully animal experimentation certified scientists according to the EU Directive 2010/63/EU and followed the Portuguese legislation for the use of laboratory animals (DL n°113/2013, 7 August). All experimental animals are younger than 5 days post-fertilization (dpf), and no ethical approval is required accordingly to Portuguese (DL n°113/ 2013, 7 August) and EU (2010/63/EU) law.

### Morpholino injection
Upon spawning, embryos were immediately collected and microinjected at the 1-2-cell stage with 1 nL of morpholino solution containing either 0.8 pmol control morpholino (CTRMO) or *mct8* morpholino (MCT8MO) (Gene Tools, USA) as described[12]. Embryos were grown on plastic Petri dishes containing E3 medium (5 mM NaCl, 0.17 mM KCl, 0.33 mM CaCl, 0.33 mM MgSO4) and reared until sampling time at 28.5 °C in an incubator (Sanyo, Germany) under 12 h:12 h light: dark cycles. Staging was done after Kimmel et al.[28] observing developmental landmarks in CTRMO embryos.

### T3 treatment
Upon spawning, embryos were immediately collected and distributed on plastic Petri dishes containing E3 medium and maintained at 28.5 °C in an incubator (Sanyo, Germany). After 10 h, the experimental group embryos were transferred to E3 medium containing 20 μM 3,3',5-Triiodo-L-Thyronine (T3, Sigma-Aldrich, USA), while the control embryos were maintained in E3 medium. Embryos were reared at 28.5 °C until sampling under 12 h:12 h light: dark cycles. The staging was done after Kimmel et al. observing developmental landmarks in control embryos[28].

### Rescue experiments
One-cell stage embryos were injected with either 0.8 pmol CTRMO, 0.8 pmol MCT8MO or 0.8 pmol MCT8MO + 100 pg *vegfaa-165* mRNA as described in ref. 12. The *vegfaa-165* gene was isolated from the pCR®4-TOPO plasmid using EcoRI (Thermo Fisher Scientific, USA) restriction enzyme according to the manufacturer's instruction and cloned into the pCS2+ plasmid. 5 μg of pCS2+ *vegfaa-165* was linearized with NotI (Thermo Fisher Scientific, USA) restriction enzyme and purified by phenol-chloroform. 1 μg of the linearized pCS2+ *vegfaa-165* was used as the template for in vitro transcription using the mMESSAGE mMACHINE® Kit (Ambion, UK) following the manufacturer's instructions. mRNA was quantified and stored at −80 °C until use.

### Analysis of gene expression
Five independent biological replicates (pools of 10 embryos) were sampled at 28, 30, 32, 36, 40, 44, 48, 54 and 72 hpf for each experimental condition (MCT8MO and CTRMO). Embryos were manually dechorionated, snap-frozen in liquid nitrogen and stored at −80 °C until use.

Total RNA extraction and cDNA synthesis were performed as described[27].

The quantification method used with the quantitative real-time PCR (qPCR) method was the absolute quantification method, which determines the number of mRNA copies in the sample from a standard curve as described in ref. 27. Primers were designed using Primer 3 Plus[29] using RNAseq data[30]. Supplementary Table 1 provides primer sequences, amplicon size and RefSeq for each gene included in the analysis. qPCR assays were run in a 384-well plate (Axygen, Life Sciences, USA) with 6 μL of reaction mixture per well (1 × SensiFAST™ SYBR, No-ROX Kit, Bioline, USA), 150 nM forward primer, 150 nM reverse primer and 1 μL cDNA (1/5)). PCR cycling condition consisted of 95 °C for 3 min, followed by 44 cycles of 95 °C for 10 s and 60 °C for 15 s. A standard melting curve analysis was included to confirm the production of a single amplicon and the absence of primer dimers that consisted of a gradient temperature increment of 0.5 °C for 5 s from 60 °C to 95 °C. A negative control was included to detect genomic DNA contamination. Total RNA input was used as a normalizer since no commonly used reference gene (18S and *gapdh*) presented invariable expression during the embryonic stages analyzed[31].

### Fluorescent immunohistochemistry (IHC)
*Tg(fli1:EGFP)*[22] experimental embryos were fixed at selected stages in ice-cold 4% paraformaldehyde (PFA)/1 × Phosphate-buffered saline (PBS) overnight at 4 °C. IHC was carried out as previously described in ref. 27. Samples were incubated overnight at 4 °C with 1:1000 rabbit anti-GFP (ab290, Abcam, UK) primary antibody and detected with 1:400 goat-anti-rabbit 488H + L (111-545-047, Jackson Laboratory, USA).

### Double fluorescent immunohistochemistry (IHC)
A double IHC procedure was performed in *Tg(gSA2AzGFF306A)*[25] and double transgenic (*Tg(gSA2AzGFF306A);Tg(kdrl:CaaX-mCherry)*)[23] experimental zebrafish embryos at 32, 36 and 48 hpf. Antibody detection and development of the signal were carried out sequentially. IHC was carried out as described previously. Primary antibodies used were rabbit anti-GFP (Abcam, UK, 1:1000). Second primary antibodies used were 1:100 mouse anti-mCherry-Tag (STJ34373, St John's Laboratory, UK), 1:100 mouse anti-Zrf1 (ZDB-ATB-081002-46, ZIRC, USA), 1:500 mouse anti-HuC/D (16A11, Invitrogen, USA). Secondary antibodies were goat-anti-rabbit 488H + L (Jackson Laboratory, USA, 1:400) and goat anti-mouse IgG-CF594 (SAB4600321, Sigma-Aldrich, USA, 1:400). Samples were stored in 1 × PBS containing 0.1% Dabco (CarlRoth, Germany).

### Colorimetric whole-mount in situ hybridization (WISH)
Riboprobe synthesis, hybridization, and imaging of colorimetric WISH were performed as described[12]. The *vegfaa* and *vegfab* plasmids were kindly provided by Brant Weinstein, the *pdgfrb* plasmid was kindly provided by Ching-Ling Lien, the *gad67a*, *gad67b*, *glyt2a* and *glyt2b* plasmids were kindly

provided by Wolfgang Driever, the *cadherin5* plasmid was kindly provided by Wiebke Herzog, the *dsRed-1* plasmid was kindly provided by Raquel Andrade, the *thraa* and *thrab* plasmids were kindly provided by Sachiko Takayama. The zebrafish *pax8* (XM 001339857.2) riboprobe was generated as described in Campinho et al. [12]. The zebrafish *pax6a* riboprobe was generated after PCR cloning against the *pax6a* transcript[30] (Supplementary Table 2). Images were acquired using an Olympus SX7X stereoscope with an Optika digital color camera.

### Double fluorescent whole-mount in situ hybridization (WISH)

Riboprobes were generated as described in the previous section and labeled with either digoxigenin (Dig) (*gad67a*, *gad67b*, *glyt2a* and *glyt2b*) or fluorescein (Fluo) (*dsRed-1*). WISH was carried out as described in ref. [27]. The signal from the first probe was generated after using the FastRed/Naphthol AS-MX substrate (Sigma-Aldrich, USA) and the signal from the second probe was generated with FITC-Tyramide (Perkin-Elmer, USA). Samples were stored until imaging in $1 \times$ PBS containing 0.1% Dabco (CarlRoth, Germany).

### Fluorescent whole-mount in situ hybridization (WISH) with immunohistochemistry (IHC)

Riboprobes were generated as described in the previous section and labelled with digoxigenin (Dig) (*pdgfrb*, *pax6a*, *pax8* and *vegfaa*). A fluorescent WISH with IHC was performed using *Tg(fli1:EGFP)*[22] or *Tg(gSA2AzGFF306A)*[25] zebrafish embryos (32, 36, 48 hpf).

The procedure was carried out as described[27], and fluorescent color development was carried out using FITC-Tyramide (Perkin-Elmer). The primary antibody used was rabbit anti-GFP (Abcam, UK, 1:500) and secondary antibody fluorescent labeling was carried out using goat anti-rabbit CF594 (Sigma-Aldrich, USA, 1:400). Samples were stored until imaging in $1 \times$ PBS containing 0.1% Dabco (CarlRoth, Germany).

### Double fluorescent whole-mount in situ hybridization (WISH) with immunohistochemistry (IHC)

Riboprobes were generated as described in the previous section and labelled with digoxigenin (Dig) (*mct8*, *thraa* and *thrab*) or fluorescein (Fluo) (*pax6a*). A double hybridization procedure combining one Dig and one Fluo probe was performed using *Tg(fli1:EGFP)*[22] zebrafish embryos (30, 32, 36, 42, 48 hpf) as described[27]. Afterwards, IHC against GFP was carried out with a rabbit anti-GFP (Abcam, UK, 1:500) anti-serum. The secondary antibody fluorescent labelling was done using goat anti-rabbit CF633 (Sigma-Aldrich, USA, 1:400). Samples were stored until imaging in $1 \times$ PBS containing 0.1% Dabco (CarlRoth, Germany).

### Evans blue staining

For live observation of hindbrain blood vessel, 48 hpf wild-type ($n = 15$) and $mct8^{(-/-)}$ ($n = 13$) embryos were incubated for 10 min at room temperature in E3 medium with 0.1% Evans blue accordingly to Eliceiri et al. [32]. After incubation, the embryos were washed five times for 5 min in an E3 medium without Evans blue. After staining, the embryos were mounted in 3% methylcellulose/E3 medium and imaged in a Zeiss Z2 microscope coupled to Hamamatsu Orca V2 digital camera.

### Generation of *pax6a* loss-of-function mutant

CRISPRScan[33] was used to design a guide RNA (gRNA) against exon 7 of the zebrafish *pax6a* locus (GGTTGAGGTTGTGCCCGAGG). gRNAs were in vitro transcribed as described[33] and purified by sodium acetate and ethanol precipitation. *pax6a* gRNA was diluted to 300 ng/μL in a 600 ng/μL Cas9 protein (Weissman Institute, Israel) and injected in 1-cell stage zebrafish *Tg(kdrl:CaaX-mCherry)*[23] embryos.

At 24 hpf, eight embryos per injection clutch were collected for genotyping by PCR. Genomic DNA extraction was carried out after overnight digestion at 50 °C in genomic extraction buffer (10 mM Tris pH8.2, 10 mM EDTA, 20 mM NaCl, 0.5% sodium dodecyl sulfate (SDS), 200 ng/mL Proteinase K), followed by centrifugation and washing with 70% ethanol, air dried, and resuspended in 20 μL of Tris-EDTA buffer (TE) pH 8. PCR was carried out with primers (0.2 μM) flanking the gRNAs binding sites (For – GATAGTGCACATTGTAGCAA; Rev – CAGCCCAGCCAGACCTCAT CC) using the DreamTaq polymerase kit (Thermo Fisher Scientific, USA). Thermocycling was carried out as follows: 5 min 95 °C, 35 cycles of 30 s 95 °C, 30 s 58 °C, 30 s 72 °C, followed by a final 5 min step at 72 °C. PCR products were fractioned in a 3.5% agarose/$1 \times$ Tris-acetate-EDTA buffer (TAE) gel. gDNA of a control wild-type embryo was used as a negative control. Bands were isolated from the gel and extracted with a gel extraction band kit (OMEGA Biotek, USA), followed by Sanger sequencing using the Big-dye termination method.

The isolated band sequence was confirmed after BLAST analysis and alignment to the zebrafish *pax6a* locus. That ensured that injected clutches had embryos carrying the desired genetic lesions on the *pax6a* locus. Injected embryos were reared until adulthood. After isolation, adult-injected PCR genotyped fish after fin-clipping to identify carriers of genetic lesions on the *pax6a* locus. After sequencing, only carriers of mutations were allowed to cross with wild-type siblings to give rise to non-mosaic F1 carrier lines. Adult *Tg(kdrl:CaaX-mCherry)*[23] F1 carriers were genotyped by PCR after fin-clipping and sequenced. In-crosses were carried out to generate F2 homozygous mutants for the *pax6a* locus.

Fluorescent live images of the hindbrain vasculature from *Tg(kdrl:CaaX-mCherry)*[23] wild-type and *pax6a*$^{(-/-)}$ were taken at 4 dpf in a Zeiss Z2 microscope coupled to Hamamatsu Orca V2 digital camera.

### Live imaging

Zebrafish *Tg(fli1:EGFP)*[22] were mated, and 1-cell stage eggs were injected with either 0.8 pmol CTRMO or MCT8MO (Gene Tools, USA) as described[12]. At 28 hpf, zebrafish embryos were screened for GFP fluorescent protein driven by the *fli1* promoter, a marker for endothelial cells. Zebrafish *Tg(kdrl:CaaX-mCherry)*[23] and wild-type AB were mated and 1-cell stage eggs were injected with *pax6a* gRNA+Cas9 (300 ng/μL + 600 ng/μL) (Weissman Institute, Israel) as described previously. At 28 hpf, zebrafish embryos were screened for the membrane-bound Cherry fluorescent protein driven by the *kdrl* promoter, a marker for endothelial cells. Afterwards, embryos were screened for positive *pax6a* mutation by selecting embryos with a small eye[34].

Imaging was carried out by light-sheet microscopy, Lightsheet Z1 (ZEISS, Germany). Briefly, embryos were anesthetized with 0.08% tricaine pH 7.4 buffered, mounted alive in 0.3% (w/v) low-melting agarose (LMA) (CarlRoth, Germany) in E3 medium containing tricaine (0.08%) into FEP tubes closed with a 1% LMA. Three animals per group, CTRMO and MCT8MO or not injected control and *pax6a* crispants, were imaged in the same tube. Two independent experiments were carried out. Time lapse images were taken from 32 until 50 hpf with a z-step of 0.740 μm to acquire the complete hindbrain and were imaged every 20 min for 20 h. The hindbrain was imaged with a $10 \times$ lens, $1 \times$ zoom with dual illumination. For image analysis, dual illumination images from the Z1 were merged using Dual side Fusion (Zen Black, Zeiss). Afterwards, images were imported into Fiji[35] and analyzed.

### Image acquisition and analysis

Fluorescent imaging of the hindbrain was carried out using a Zeiss Z2 microscope coupled to a Zeiss digital camera or a Lighsheet Z1 microscope (Zeiss, Germany). Samples for Z2 imaging were mounted in 0.3% agarose (Sigma-Aldrich, USA) and imaged using a $20 \times$ lens and a z-step of 0.850 μm. Afterwards, images were deconvoluted in SCI Huygens software 4.4 (Scientific Volume Imaging, The Netherlands, http://svi.nl). Maximum projections were generated in Fiji[35]. Samples for Z1 imaging were mounted in 1% Low melting agarose (CarlRoth, Germany) and imaged using a $10 \times$ lens, dual illumination and a z-step of 1.5 μm (optimal distance option) to acquire the complete hindbrain using $1 \times$ zoom. After image acquisition, dual illumination images were merged using Dual side Fusion (Zen Black, Zeiss, Germany). Images were then imported into Fiji[35] and analyzed.

Colocalization Colormap plugin[36] was used to determine the colocalization of Dig and Fluo cRNA probes.

## Statistical analysis

All statistical analyses were done in GraphPad Prism version 8.4.0 software for Mac (San Diego, USA, www.graphpad.com). Values are represented as means ± SD. The levels of statistical significance were expressed as p-values, *$p < 0.05$; **$p < 0.01$; ***$p < 0.001$; ns: non-significant. Due to the role played by the genes analyzed in embryonic development, the present work did not intend to determine their temporal expression patterns, only the effect of *mct8* knockdown on their expression at specific time points. To determine gene expression differences between CTRMO and MCT8MO embryos, statistical significance was determined by unpaired Students *t*-test: two-sample, assuming equal variances. One-way analysis of variance (ANOVA) or Fisher's exact test was used to determine statistically significant differences ($p < 0.05$) between control and experimental embryos.

## Reporting summary

Further information on research design is available in the Nature Portfolio Reporting Summary linked to this article.

## Results

### Zebrafish hindbrain vasculature is altered in *mct8* morphants and loss of function mutants

A previous study showed that the *mct8* morphant zebrafish embryo at 48 hpf presents an impaired hindbrain vasculature development, displaying only 3 of the 7 central arteries (CtAs) (Supplementary Fig 1a)[12]. These results were further confirmed in the recently developed *mct8* knockout mutant (*mct8*[(-/-)])[27] using Evans blue staining (Supplementary Fig 1b). This evidence shows that MT3 is involved, via Mct8 transport, in CtAs and hindbrain vasculature development.

### MT3 action during zebrafish hindbrain vasculature development

Previous transcriptomic analysis between control morpholino (CTRMO) and *mct8* morpholino (MCT8MO) zebrafish embryos at 25 hpf, showed that before hindbrain vasculature development, angiogenic genes are differentially expressed (Supplementary Fig 2a, $p < 0.01$; FDR < 0.05)[30]. We extended gene expression analysis to a time window from 28 to 72 hpf (Supplementary Fig 2b) that encompasses the development of the hindbrain vasculature[37]. Despite different expression dispersion, whole-embryo quantitative real-time PCR (qPCR) gene expression analysis also did not find significantly different expression between CTRMO and MCT8MO embryos from 28 to 72hpf (Supplementary Fig 2b–d).

Although CtAs development is affected in MCT8MO embryos, this does not occur in intersegmental trunk vessels, indicating that potential MT3 action on CtAs is specific to the brain (Supplementary Fig 1a). Moreover, neither *mct8* nor T3 receptors *thyroid hormone receptor aa* (*thraa*), *thyroid hormone receptor ab* (*thrab*), and *thyroid hormone receptor b* (*thrb*) are expressed in endothelial cells before 48 hpf[12], suggesting an indirect effect on CtAs. Nonetheless, given the vascular hindbrain phenotype, we decided to analyze the in situ expression of the *vascular endothelial growth factor a* (*vegfa*) ligands given their essential role in zebrafish CtA development[38]. Whole-mount in situ hybridization (WISH) analysis revealed a decrease in *vegfaa* hindbrain expression at 36 hpf in MCT8MO embryos (Supplementary Fig 3a) and in a region where CtAs are expected to sprout (green arrowheads in Supplementary Fig 3a). WISH analysis for *vegfab* revealed a more widespread expression in the brain than *vegfaa* (Supplementary Fig 3). The expression pattern of *vegfab* between CTRMO and MCT8MO zebrafish embryos was mostly maintained. Still, the signal was less intense in MCT8MO than in CTRMO zebrafish embryos (Supplementary Fig 3b). The results suggest that MT3 is required for appropriate *vegfaa* and, to some extent, *vegfab* expression in the hindbrain.

Given that *vegfaa* is the principal angiogenic factor involved in CtA development[38], we analyzed the spatial relationship between hindbrain *vegfaa* expression and the position of CtA ingression using the *Tg(fli1:EGFP)* transgenic zebrafish line (Fig. 1a). In CTRMO embryos expression of *vegfaa* at 32, 36 and 48 hpf can be observed juxtaposed to specific CtAs, showing the importance of *vegfaa* for CtA sprouting and migration. In MCT8MO embryos, *vegfaa* hindbrain expression was not lost, but significantly reduced (Fig. 1a–d). At 32 hpf, the knockdown of *mct8* significantly reduced the expression of *vegfaa* in rhombomeres 2, 3, and 5 (Fig. 1b); at 36 hpf, it reduced considerably in rhombomeres 2, 3, and 4 (Fig. 1c); at 48 hpf, it was significantly reduced in rhombomeres 1 to 5 (Fig. 1d). These results argue that MT3 regulates *vegfaa* hindbrain expression during CtA development.

To functionally determine if MT3 acts upstream of *vegfaa* during hindbrain vasculature development, we co-injected zebrafish *vegfaa-165* mRNA with MCT8MO (Fig. 1e). Embryos injected with CTRMO have 5 to 7 CtAs present, as expected at this developmental stage[37] (Fig. 1f). The number of CtAs is significantly reduced in the MCT8MO embryos, with an average of 2 CtAs present at 48 hpf (Fig. 1f). Analyzing the frequency of each CtA developed between CTRMO and MCT8MO embryos shows that all 7 CtAs were significantly affected (Fig. 1g). Exogenous supply of *vegfaa-165* mRNA significantly increases the number of CtAs to an average of 4 CtAs compared to MCT8MO alone (Fig. 1f), being able to rescue CtAs 1, 2, 3 and 7 (Fig. 1g). The embryos co-injected with MCT8MO and *vegfaa-165* mRNA partially rescued the phenotype (Fig. 1e, f) even though these were still significantly different. CtAs 5 and 6 were partially recovered by the exogenous supply of *vegfaa-165* mRNA, but CtA 4 cannot be recovered (Fig. 1g). This evidence argues that for these CtAs *vegfaa* is likely not the only signal under the regulation of MT3 required for ingression into the hindbrain. Noteworthy, the ingression of CtAs into the hindbrain in co-injected MCT8MO+*vegfaa-165* mRNA was not identical to CTRMO embryos (Fig. 1e), and these present a shallow ingression in the hindbrain. These can be due to impaired CtA development after ingression or a consequence of altered hindbrain development[27,30], or both.

We further investigated if the impaired development of the hindbrain vasculature could give rise to changes in the neurovascular unit that is formed. To determine the involvement of MT3 in establishing a functional blood-hindbrain-barrier (BHB) in zebrafish we used brain pericyte recruitment for hindbrain vasculature as a proxy. We quantified the number of pericytes following WISH for *platelet-derived growth factor receptor-b* (*pdgfrb*) after T3 supplementation and in *mct8* knockdown (Fig. 2a, b). Treatment with T3, compared with control embryos, did not affect the recruitment of pericytes to the hindbrain vasculature (Fig. 2a). However, compared to CTRMO, in *mct8* morphants, there was a delay in the pericyte presence on the primordial hindbrain channels (PHCB) and basilar artery (BA) (Fig. 2b). In the PHBC of MCT8MO embryos, the pericyte number was significantly reduced at 48 hpf (Fig. 2f). In the BA, the pericyte number increased significantly at 36 hpf and then decreased significantly at 48 hpf (Fig. 2g, h). Interestingly, at 36 hpf, the BA in the MCT8MO was not yet completely formed, and the pericytes formed agglomerates in the posterior region of the BA (Fig. 2b). Nevertheless, the BA was completely formed at 48 hpf in MCT8MO embryos but presented fewer pericytes (Fig. 2b, g). This observation strongly suggests that impaired pericyte migration in MCT8MO embryos is not due to direct effects on these cells but due to delayed BA development. Pericytes in CtAs were only observed at 48 hpf in CTRMO embryos, while in MCT8MO embryos, no pericytes were observed (Fig. 2b, h). Since there is no increase in pericyte number after adding T3 to the medium, we suggest that MT3 does not act directly on pericytes, but instead, the delayed/impaired vascular development seems to affect the number of pericytes being recruited to these vascular structures.

### *pax8* and *cpne4* hindbrain neurons are under MT3 regulation

The previous results have shown that the angiogenic factor *vegfaa* is under MT3 regulation and thus responsible for CtA development. However, the question is, which neural cells depend on MT3 signaling and are thus responsible for *vegfaa* expression? Previous zebrafish spinal cord vascularization studies have shown that neurons coordinate intersegmental vessel development by expressing neuron-derived *vegfaa* and *soluble fms-like*

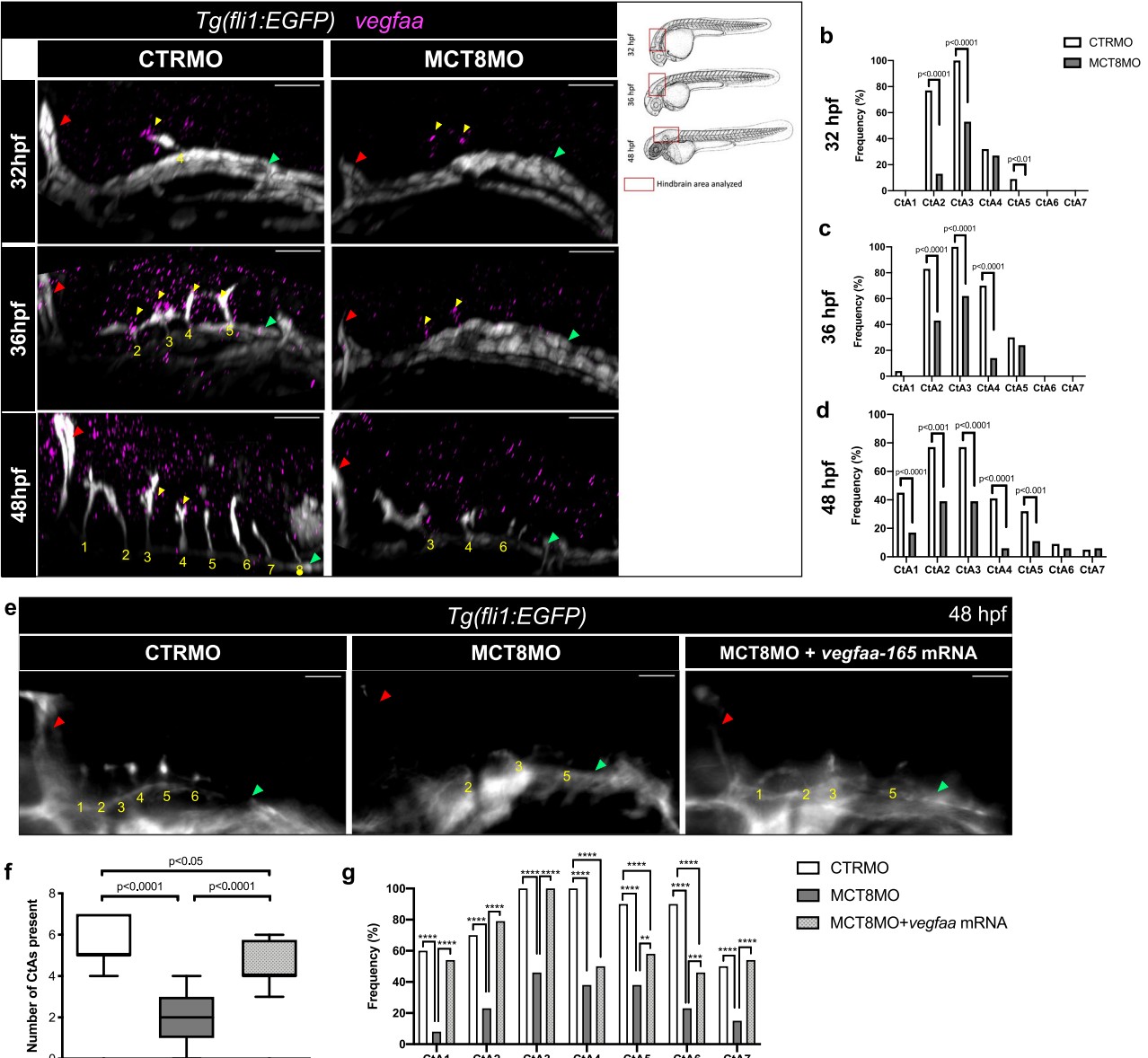

**Fig. 1 | MT3 regulates *vegfaa* expression during hindbrain vascular development.**
**a** Lateral maximum projections of the hindbrain of *Tg(fli1:EGFP)* after WISH
against *vegfaa* (magenta) and immunostained against GFP (endothelial marker,
white) in CTRMO and MCT8MO zebrafish embryos at 32, 36 and 48 hpf. Yellow
arrowhead indicates agglomeration of *vegfaa* expression around a CtA or the
putative location where a CtA is supposed to ingress into the hindbrain. The red box
indicates the hindbrain area for every embryonic stage analyzed adapted from
(Kimmel et al., 1995). Graphical representation of the number of times, defined as
frequency (%), that *vegfaa* expression can be observed around a specific CtA or
where a CtA is supposed to develop at **b**) 32 hpf, **c**) 36 hpf, **d**) 48 hpf between
CTRMO and MCT8MO embryos; Fisher's exact test.*n* = 22 (32hpf and 48hpf
CTRMO), 15 (32hpf MCT8MO), 23 (36hpf CTRMO), 21 (36hpf MCT8MO), 18
(48hpf MCT8MO). **e** At 48 hpf, *Tg(fli1:EGFP)* zebrafish embryos co-injected with
*vegfaa-165* mRNA and MCT8MO (*n* = 24) are able to partially rescue, in a

rhombomere specific manner, some but not all CtAs. CTRMO (*n* = 10), MCT8MO
(*n* = 14). **f** Statistical analysis of the number of CtAs present in each experimental
group; One-way ANOVA followed by Bonferroni's multiple comparison analysis.
Data are presented as box-and-whisker plot, where the black thick horizontal line
represents the median. The first and third quartiles are marked by the lower and
upper edges of the boxes, respectively. Error bars represent standard deviations
(smallest and higher value). **g** Frequency (%) of each CtA to develop in each
experimental group; Fisher's exact test. **p < 0.01; ***p < 0.001; ****p < 0.0001.
For detailed statistics, see Supplementary Data 2. The red arrowhead indicates the
mid-cerebral vein (MCeV), and the green arrowhead indicates the primordial
hindbrain channels (PHBC). Numbers in yellow 1 – 8 indicate the CtA in its
respective rhombomere. Scale bar: 50 μm. Zebrafish drawings used from Kimmel
et al (1995)(Ref. 28) under CC licence number 6041951178663.

*tyrosine kinase 1 (sflt1)*[39]. One hindbrain population previously shown to be
under MTH influence are *pax8* inhibitory interneurons[12]. We explored the
spatial relationship between *pax8* expression and CtA development. In
CTRMO embryos, *pax8* expression at 32 and 36 hpf was located in the
posterior region of the hindbrain and the midbrain-hindbrain boundary
(MHB) (Fig. 3a). At 48 hpf, *pax8* was expressed in the hindbrain juxtaposed
to CtAs 4 to 7 (Fig. 3a, b). In MCT8MO embryos, some *pax8* expression was
detected at 32 and 36 hpf in the MHB, but in the hindbrain, only at 48 hpf

*pax8* expression was present but in a more reduced expression field
(Fig. 3a, c). Most of the remaining *pax8*-expressing cells were located more
ventrally in MCT8MO embryos than in CTRMO embryos, suggesting a
spatial redistribution of these cells and likely a different identity and function
(Fig. 3b, c). This observation suggests a relation between the lack of or
reduced *pax8* expression in the posterior region of the hindbrain in
MCT8MO zebrafish embryos and the lack of sprouting CtAs from rhom-
bomeres 4 to 7. To understand if *pax8*-neurons are involved in CtA

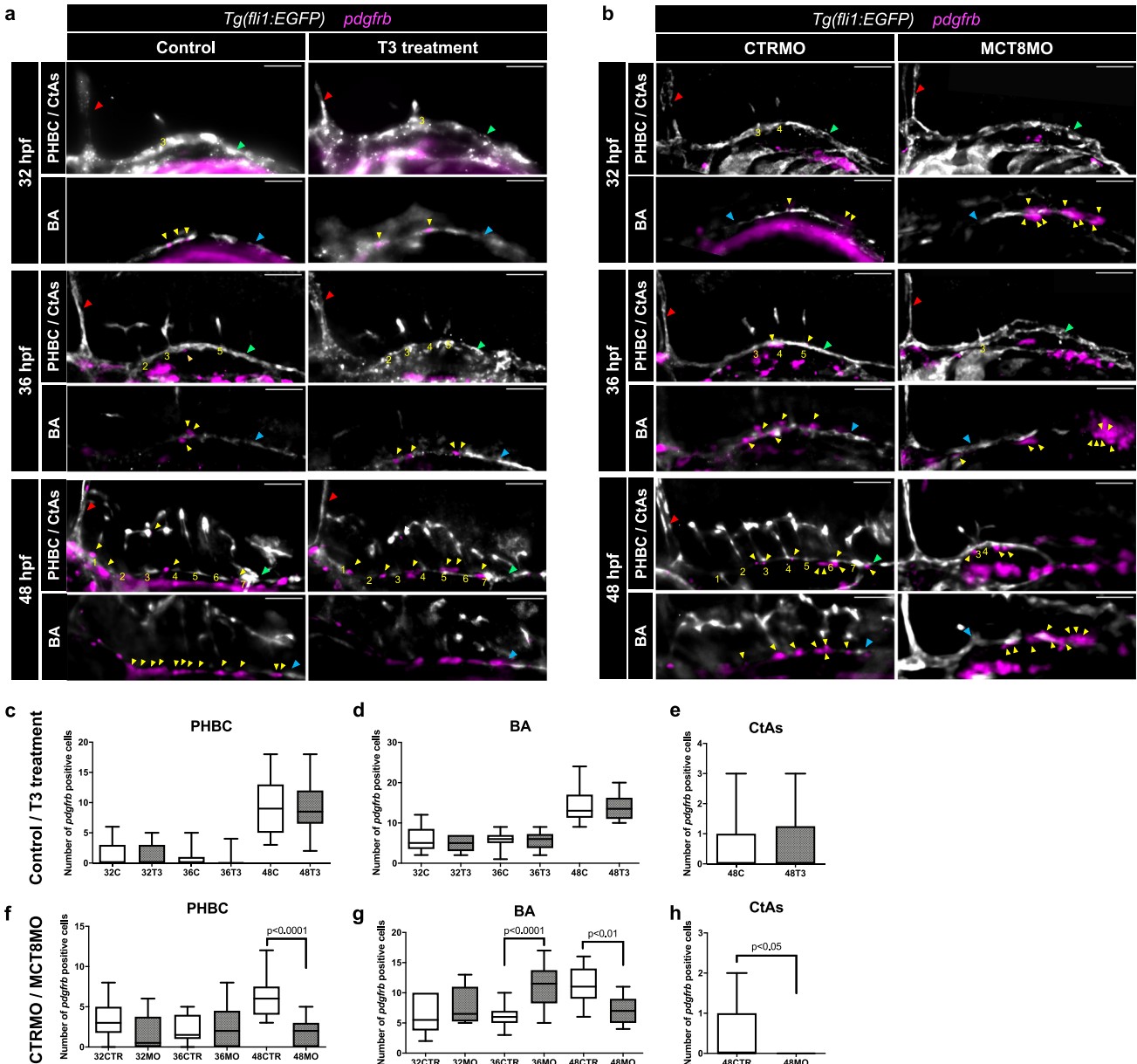

**Fig. 2 | MT3 seems to affect indirectly pericyte number due to impaired development of the hindbrain vascular structures.** Lateral view of *Tg(fli1:EGFP)* zebrafish embryos after WISH for *pdgfrb* (pericyte marker, magenta) and immunostained against GFP (endothelial marker, white) at 32, 36 and 48 hpf. Zebrafish embryos were submitted to two experimental conditions: **a** Increase of T3 (20 mM) availability in the medium (Control vs. T3 treatment) and **b** Knockdown of the Mct8 transporter by a morpholino-based system (CTRMO vs. MCT8MO). The red arrowhead indicates the mid-cerebral vein (MCeV), the green arrowhead indicates the primordial hindbrain channels (PHBC), the blue arrowhead indicates the basilar artery (BA), and the yellow arrowhead indicates pericytes. Yellow numbers 1-7 indicate the central arteries (CtAs) in their respective rhombomere.

Scale bars: 100 μm. Quantification of the pericyte numbers in Control and T3 treatment condition in the **c** PHBC, **d** BA, and **e** CtAs at 32 hpf (*n* = 13, 11 (C, T3)), 36 hpf (*n* = 17, 14 (C, T3)) and 48 hpf (*n* = 20, 18 (C, T3)). Quantification of the pericyte numbers in CTRMO and MCT8MO condition in the **f**) PHBC, **g**) BA, and **h**) CtAs at 32 hpf (*n* = 14, 12 (CTRMO, MCT8MO)), 36 hpf (*n* = 15, 12 (CTRMO, MCT8MO)) and 48 hpf (*n* = 16, 11 (CTRMO, MCT8MO)). Data are presented as box-and-whisker plot, where the black thick horizontal line represents the median. The first and third quartiles are marked by the lower and upper edges of the boxes, respectively. Error bars represent standard deviations (smallest and highest value). Statistical significance determined by *t*-test *p* < 0.05. For detailed statistics, see Supplementary Data 2.

development, we used a homozygous transgenic *Tg(pax8:DsRed)* zebrafish line that presents *pax8* loss of function[24]. WISH analysis for the vascular endothelial cadherin (*VE-cadherin* or *cadherin-5*) between heterozygous *pax8*[+/-] (control) and hypomorph *pax8*[-/-] embryos, which is specifically expressed in endothelial cells, shows that *pax8*[-/-] hypomorph mutants develop a normal hindbrain vasculature (Fig. 3d). Comparative analysis of the length of the PHBC (Fig. 3f) and CtAs (Fig. 3e) between control *pax8*[+/-] and hypomorph *pax8*[-/-] mutant embryos showed no significant differences between them, except for CtA 7, usually the last to develop[37], which can lead

to its length variance at 48 hpf. These results demonstrate that *pax8*-neurons are not involved in CtA ingression.

Since *pax8* gene function is lost in hypomorph *pax8*[-/-] mutant embryos and they developed a normal BHB, this raises the question of whether other glycinergic (*glyt2a* and *glyt2b* positive) and GABAergic (*gad67a* and *gad67b* positive) interneurons (from now on called "inhibitory") in the hindbrain can be behind CtA ingression. In control, pax8[+/-] embryos, colocalization between DsRed expressing cells and inhibitory cells can be observed, while in hypomorph *pax8*[-/-] mutant zebrafish embryos, most of this colocalization

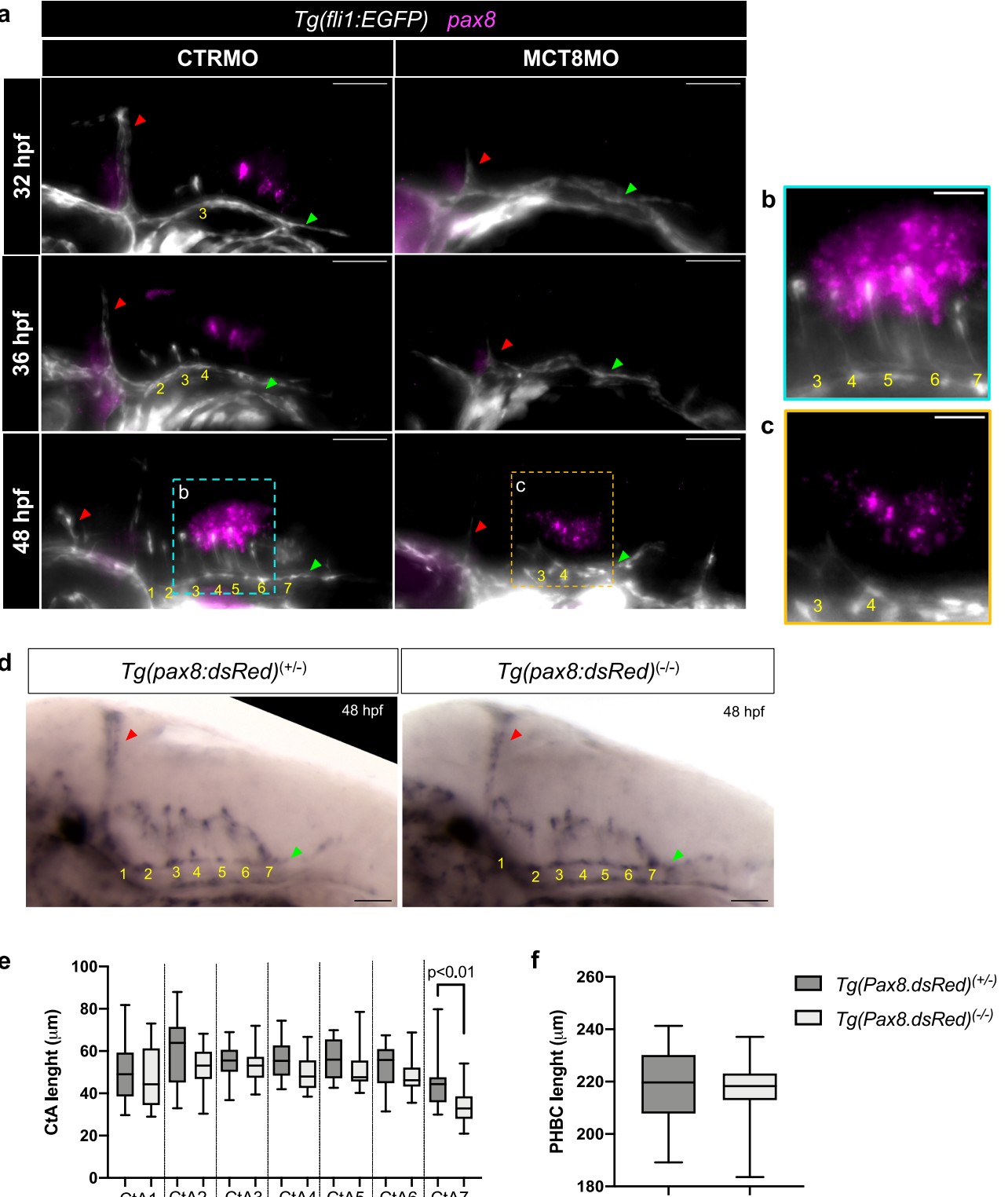

**Fig. 3 | *pax8* expression is downregulated in MCT8MO zebrafish embryos but is not involved in hindbrain vascular development. a** Lateral view of maximum projections of the hindbrain vasculature structures at 32, 36 and 48 hpf in *Tg(fli1:EGFP)* in CTRMO and MCT8MO zebrafish embryos after WISH for *pax8* (magenta) and immunostaining for GFP (endothelial marker, white). In CTRMO zebrafish embryos, *pax8* expression appears in the posterior hindbrain region, juxtaposed to CtAs 4 to 7. In MCT8MO zebrafish embryos, *pax8* expression was clearly reduced and only appears at 48 hpf (*n* = 9 – 18). Magnification of the selected area at 48 hpf for **b** CTRMO and **c** MCT8MO embryos are shown. **d** Lateral view of the hindbrain of *Tg(pax8:dsRed)*[+/-] (control group (*pax8*[+/-])) and *Tg(pax8:dsRed)*[-/-] (hypomorph group (*pax8*[-/-])) zebrafish embryos after WISH for *cadherin-5* at 48 hpf. All 7

CtAs were present in control *pax8*[+/-] and hypomorph *pax8*[-/-] zebrafish embryos. Image J software was used to measure the length of **e** each CtA and **f** the PHBC between control *pax8*[+/-] (*n* = 18) and hypomorph *pax8*[-/-] (*n* = 14) zebrafish embryos. No changes were observed; Unpaired t-test (Mann-Whitney test). Data are presented as box-and-whisker plot, where the black thick horizontal line represents the median. The first and third quartiles are marked by the lower and upper edges of the boxes, respectively. Error bars represent standard deviations (smallest and highest value). For detailed statistics, see Supplementary Data 2. The red arrowhead indicates the mid-cerebral vein (MCeV), and the green arrowhead indicates the primordial hindbrain channels (PHBC). Yellow numbers 1 – 7 indicate the developed CtA in its respective rhombomere. Scale bar 50 μm for figures (**a**) and (**d**) and 20 μm for (**b**) and (**c**).

was lost (Supplementary Fig 4a). That indicates that the remaining DsRed cells in hypomorph $pax8^{-/-}$ mutants are likely not neurons. We used the confined displacement algorithm (CDA) for colocalization significance analysis to compute the Pearson correlation coefficient between control $pax8^{+/-}$ and hypomorph $pax8^{-/-}$ mutant zebrafish embryos. The distribution of correlation coefficients in the control $pax8^{+/-}$ group follows a weak positive correlation [0.0107– 0.1280], while the hypomorph $pax8^{-/-}$ mutant group follows a weak negative correlation [0.0432–−0.1503] (Supplementary Fig 4b). This result shows that the inhibitory cell fate in hypomorph $pax8^{-/-}$ mutant embryos has diverged from that of the control $pax8^{+/-}$ zebrafish embryos. These results indicate that, although the inhibitory cell fate may have changed, it does not seem to affect CtA development, indicating that these cells are not responsible for CtA ingression into the hindbrain.

Given that $pax8$-neurons are not involved in CtAs ingression, we argued that another neuron population might be responsible for instructing BHB development. We identified the $Tg(gSA2AzGFF306A)$ line, which presents a discrete population of hindbrain GFP(+) cells located dorsally juxtaposed and perfectly aligned along the PHBC. Genomic data from these fish indicated that the enhancer trap construct containing GFP locates in the $cpne4$ locus (https://shigen.nig.ac.jp/zebrafish/strain/strainTopJa.jsp)[25]. In mice, $Cpne4$ is an embryonically expressed gene labeling specifically the intrinsic digit-innervating excitatory glutamatergic motor neurons[40–43]. In mice, blood vessel development in the spinal cord is promoted by motor neurons by a combination of attractive and repelling signaling cues[44,45]. So, we argued if these Cpne4 cells could constitute the neuronal cell population responsible for CtAs ingression. We analyzed the hindbrain of double transgenic zebrafish embryos ($Tg(gSA2AzGFF306A;Tg(kdrl:CaaX-mCherry))$) where Cpne4/GFP(+) cells were located dorsally to the PHBCs and juxtaposed to the developing CtAs in CTRMO embryos (Fig. 4a). It was visible that the number of Cpne4/GFP(+) cells increases over development in CTRMO embryos (Fig. 4a–d). In MCT8MO embryos at 36 hpf, the number of Cpne4/GFP(+) cells was significantly reduced in rhombomeres 5 and 7, compared to CTRMO embryos (Fig. 4c). That was also evident at 48 hpf, where the number of Cpne4/GFP(+) cells was significantly reduced in rhombomeres 2, 4, 5, 6 and 7 (Fig. 4d). Given the close location of these cells with the CtAs and the fact that these cells were significantly reduced in MCT8MO embryos, the correlation between the presence of Cpne4/GFP(+) cells and the presence of CtAs in CTRMO and MCT8MO at 48 hpf was analyzed. The data showed a positive correlation between Cpne4/GFP(+) cells and the development of CtAs 4, 6 and 7 in the hindbrain (Fig. 4e). We analyzed $vegfaa$ expression in $Tg(gSA2AzGFF306A)$ control embryos. However, no colocalization was found between Cpne4/GFP(+) cells and $vegfaa$ (Fig. 4f), arguing that these cells do not instruct CtA ingression into the hindbrain.

Together, this evidence argues that neurons were likely not the neuronal cells instructing the ingression of CtAs into the hindbrain.

### $pax6a$ NPCs are under MT3 regulation and express $vegfaa$

Previously, we found that a discreet hindbrain population of ventral $pax6a$ NPCs depend on MT3[30], arguing that these cells could be responsible for CtA development. We analyzed the expression pattern of $pax6a$ during hindbrain vascular development in CTRMO and MCT8MO zebrafish embryos (Fig. 5a). In CTRMO zebrafish embryos, $pax6a+$ cells were found along the hindbrain, and the ventral $pax6a+$ cells were near the developing CtAs (Fig. 5a). In MCT8MO embryos, $pax6a+$ cells were less abundant, and at 36 and 48 hpf the ventral $pax6a+$ cells were lost (Fig. 5a).

Given these findings, we analyzed if these $pax6a+$ hindbrain cells express $vegfaa$ and are thus responsible for CtA sprouting into the zebrafish hindbrain. Colocalization analysis, using the image J plugin Colormap[36] showed that some hindbrain $pax6a+$ cells express $vegfaa$ (Fig. 5b). This was evident in the time course analyzed, 30, 32, 36, 42 and 48 hpf. Analyzing the surrounding area of every CtA in every stage shows that colocalizing $pax6a/vegfaa$ cells occurs near or dorsal to the developing CtAs (Supplementary Fig 5, individual images of the CtAs). Notable was the observation that the developing CtAs seem to migrate towards these $pax6a/vegfaa$ colocalizing cells (individual images, Supplementary Fig 5c, e, g and i), indicating that

these cells might be responsible for the chemoattraction of the developing CtAs. We also analyzed the relationship between the developing CtAs and the co-expressing $pax6a/vegfaa$ cells between CTRMO and MCT8MO zebrafish embryos (Fig. 5c–i). The statistical analysis revealed no relationship between CtA 1 development and $pax6a/vegfaa$ co-expressing cells in the analyzed time points (Fig. 5c). In rhombomere 2, $pax6a/vegfaa$ co-expressing cells were present from 30 hpf onwards in CTRMO embryos, while these cells were absent in MCT8MO; at 48 hpf only 30% of the embryos developed CtA 2. The statistical analysis argues that the existence of $pax6a/vegfaa$ co-expressing cells might favor the development of CtA 2 (Fig. 5d). CtA 3 developed in most CTRMO and MCT8MO zebrafish embryos. However, this CtA developed later in MCT8MO embryos, at 36 hpf. This explains the statistical significance observed at 32 hpf (Fig. 5e). This strengthens the fact that $pax6a/vegfaa$ co-expressing cells in this rhombomere lead to the successful development of CtA 3 in MCT8MO embryos. CtA 4 is one of the first CtAs to develop[37], but in MCT8MO zebrafish embryos, only at 42 hpf 10% of the embryos analyzed had developed this CtA. In MCT8MO zebrafish embryos in which CtA 4 developed, $pax6a/vegfaa$ co-expressing cells were always present. This evidence shows a dependency of CtA 4 development on $pax6a/vegfaa$ co-expressing cells (Fisher's exact test $p < 0.05$, Fig. 5f). In rhombomere 5, $pax6a/vegfaa$ co-expressing cells were present in CTRMO from 30 hpf and in MCT8MO embryos but in lesser quantity. Between 32 and 42 hpf, the presence of CtAs and $pax6a/vegfaa$ co-expressing cells between CTRMO and MCT8MO embryos differed significantly, showing that $pax6a/vegfaa$ co-expressing cells are needed for CtA development (Fig. 5g). In rhombomere 6, $pax6a/vegfaa$ co-expressing cells were present in CTRMO and MCT8MO zebrafish embryos from 30 hpf. There is a dependency between CtA 6 development and $pax6a/vegfaa$ co-expressing cells at 36 and 42 hpf (Fisher's exact test, $p < 0.05$). In rhombomere 7 of CTRMO zebrafish embryos, $pax6a/vegfaa$ co-expressing cells were present early during hindbrain vascular development, while in MCT8MO embryos, $pax6a/vegfaa$ co-expressing cells were only present from 42 hpf. At 42 hpf, the statistical analysis revealed a significant positive relationship between $pax6a/vegfaa$ co-expressing cells and CtA 7 development (Fig. 5i). In summary, $pax6a/vegfaa$ co-expressing cells are likely required to develop CtAs 2, 4, 5 and 7.

### $pax6a/vegfaa$ co-expressing cells are essential for CtA migration

We investigated whether the presence of $pax6a/vegfaa$ co-expressing cells influences the direction of the growing CtA in the hindbrain. For that, the angle of the developing CtA was measured using the PHBC as origin (0 degrees). The angle was measured until the directional change of the CtA towards the BA or its ipsilateral CtA neighbor. In CTRMO zebrafish embryos, CtA 1 developed later in development (42 hpf; Fig. 6a), and this CtA was not attracted by the $pax6a/vegfaa$ co-expressing cells, mirroring the previous results (Fig. 5c). The growth angle between CTRMO and MCT8MO zebrafish embryos differed significantly because only 10% of the MCT8MO zebrafish embryos developed CtA 1 at 48 hpf (Fig. 6a). In CTRMO embryos, CtAs 2 to 5 grew towards the $pax6a/vegfaa$ co-expressing cells (Fig. 6b-e). However, mostly at 42 and/or 48 hpf, these CtAs changed the migration direction (indicated by the grey box in Fig. 6b-e), thus not being attracted anymore by the $pax6a/vegfaa$ co-expressing cells (Fig. 6b–e, grey boxes). In rhombomeres 6 and 7 $pax6a/vegfaa$ co-expressing cells were absent in CTRMO at 48 hpf (Fig. 6f, g). The analysis showed that CtAs 2, 4, 5, 6 and 7 migration angles differed significantly between CTRMO and MCT8MO embryos (Fig. 6b, d–g). The only exception was for CtA 3 (Fig. 6c). In MCT8MO embryos, CtA 3 developed after 32 hpf. Although this CtA developed later than the control group, it grew towards the direction of the $pax6a/vegfaa$ co-expressing cells until 48 hpf (Fig. 6c, h).

### Hindbrain $pax6a$ NPCs are regulated by MT3 in a cell-autonomous way

The previous results indicated that MT3 signaling is important for developing a specific population of ventral $pax6a +$ NPCs likely involved in CtA ingression into the hindbrain. T3 cellular signaling genes are expressed in

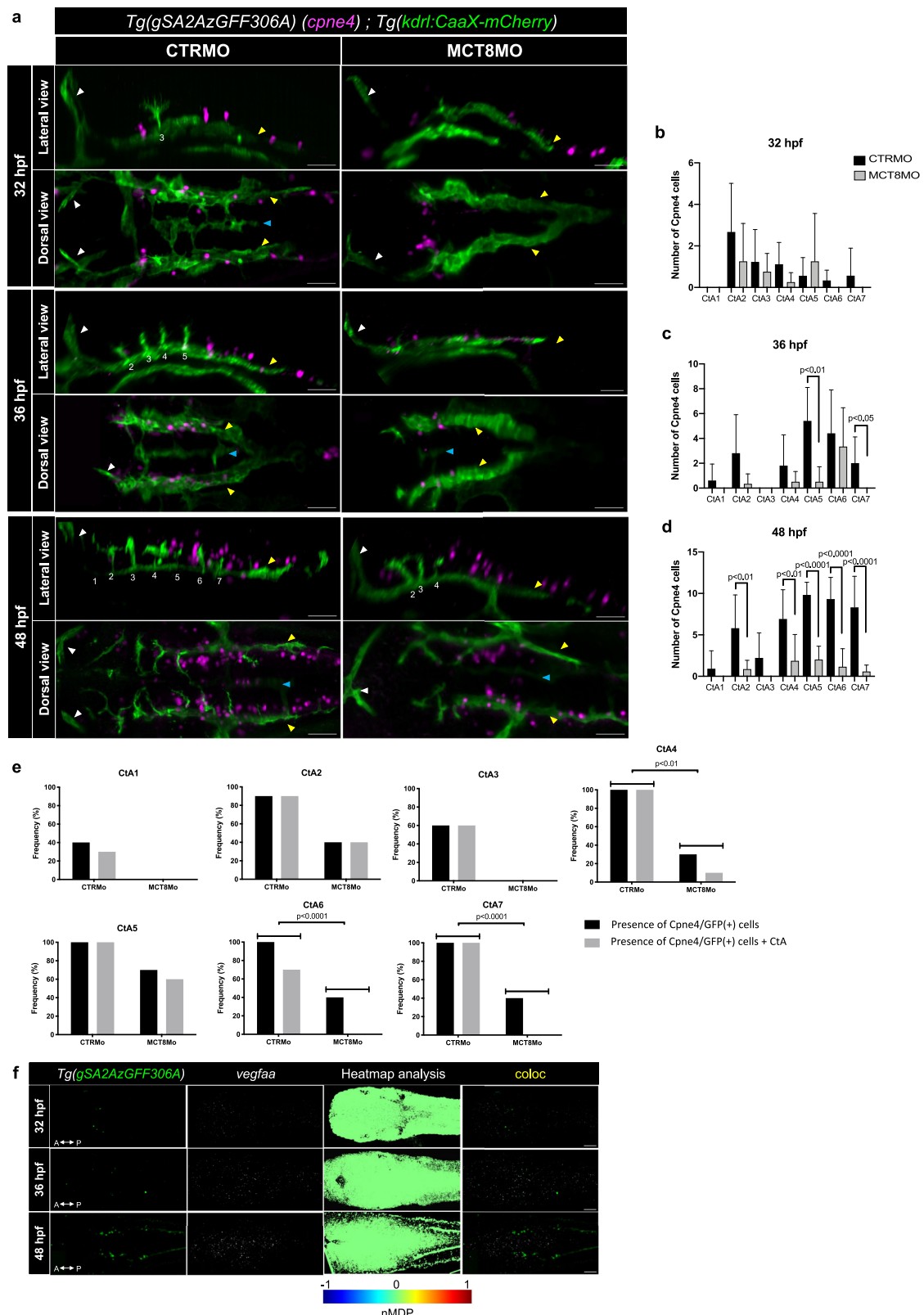

NPC of the developing cerebral cortex of mice[46], and *mct8, thraa,* and *thrab* are already expressed in the zebrafish neuroepithelium at 12 hpf [12]. From 30 hpf onwards, several MT3-sensitive *pax6a* + NPCs are present in the hindbrain of CTRMO zebrafish embryos from rhombomeres 3 to 7 but are lost in *mct8* morphants *pax6a* + MT3 sensitive (*pax6a/mct8* + ) (Fig. 7). Likewise, ventral hindbrain *pax6a*+ cell loss is also observed in *mct8*[(-/-)] loss of function homozygous embryos (Supplementary Fig 1c).

CtA 4, 5, and 6 ingression coincides or is preceded by the development of *pax6* + MT3 sensitive cell (*pax6a/mct8* + ; Fig. 7) with the ability to respond to MT3 (*pax6a/thraa/thrab* + ; Fig. 8, Supplementary Fig 6 and 7). This evidence shows direct MT3 signaling is involved in CtA 4, 5, and 6 ingressions.

CtA 1 and 2 develop independently of MT3 signaling, given that no *mct8/pax6* + cells coincide with CtA ingression (Fig. 7). In rhombomeres 1

**Fig. 4 | MT3 regulates Cpne4+ cells during hindbrain vasculature development.**
**a** Maximum projection of the hindbrain vasculature structures in double transgenic zebrafish embryos (*Tg(kdrl:CaaX-mCherry);Tg(gSA2AzGFF306A)*) immunostained against mCherry (green) and GFP (Cpne4+ cells, magenta) are shown. In CTRMO zebrafish embryos, Cpne4/GFP(+) cells were in close contact with the central arteries (CtAs), while in MCT8MO zebrafish embryos some Cpne4/GFP(+) cells were lost. The white arrowhead indicates the mid-cerebral vein (MCeV), the yellow arrowhead indicates the primordial hindbrain channels (PHBC), and the blue arrowhead indicates the basilar artery (BA). White numbers 1 – 7 indicate the developed CtA in its respective rhombomere. The number of Cpne4/GFP(+) cells between CTRMO and MCT8MO zebrafish embryos was analyzed at **b**) 32 hpf (*n* = 9, 8 (CTRMO, MCT8MO)), **c**) 36 hpf (*n* = 5, 6 (CTRMO, MCT8MO)) and **d**) 48 hpf

(*n* = 7, 10 (CTRMO, MCT8MO)); Unpaired t-test; Error bars represent standard deviation. **e** Graphical view of the correlation between the presence of Cpne4/GFP(+) cells and the presence of each CtA in CTRMO and MCT8MO at 48 hpf shows that a correlation between these two conditions exists for CtAs 4, 6 and 7. Fisher's exact test. *n* = 7 (CTRMO), 11 (MCT8MO). For detailed statistics, see Supplementary Data 2. **f** Dorsal view of the hindbrain of *Tg(gSA2AzGFF306A)* zebrafish embryos at 32, 36 and 48 hpf after WISH for *vegfaa* (white) and immunostained against GFP (Cpne4 cells, green) are shown. Heatmap colocalization analysis was used using normalized mean deviation product (nMDP) values and a combined image between GPF and *vegfaa* expression is shown. Color bar chart indicating no colocalization (-1, blue color) to colocalization (1, red color). Scale bar: 50 μm.

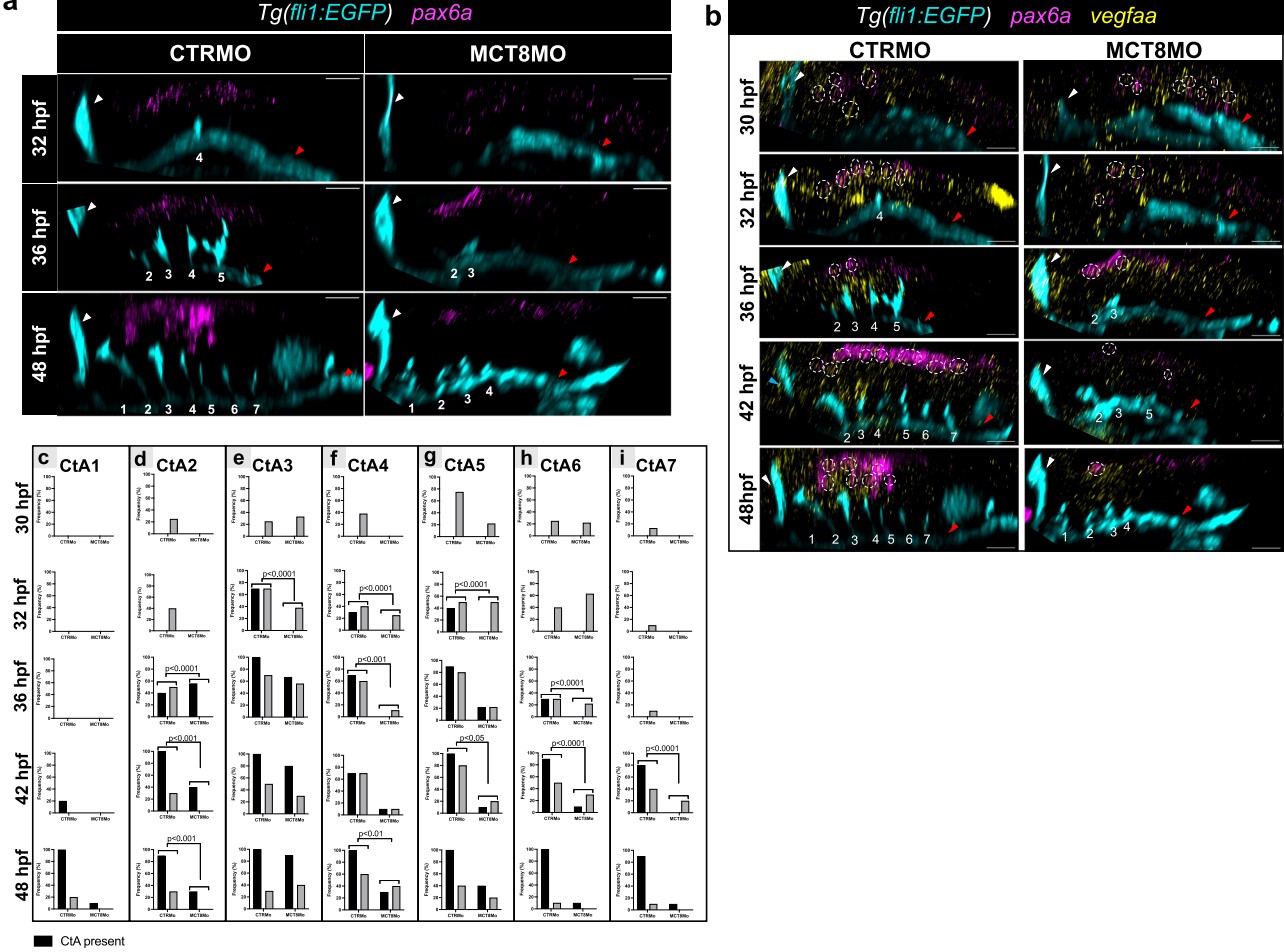

**Fig. 5 | Ventral *pax6a*+ cells express *vegfaa* in the zebrafish hindbrain. a**Lateral view of *Tg(fli1:EGFP)* CTRMO and MCT8MO zebrafish embryos after WISH for *pax6a* (magenta) and immunostained against GFP (endothelial marker, cyan) at 32, 36 and 48 hpf are shown. The ventral population of *pax6a* expressing cells were lost in MCT8MO zebrafish embryos, compared to CTRMO embryos. **b**Lateral view of *Tg(fli1:EGFP)* fluorescent maximum projection images of double WISH for *pax6a* (magenta) and *vegfaa* (yellow) and immunostained against GFP (cyan) in CTRMO and MCT8MO zebrafish embryos at 30, 32, 36, 42 and 48 hpf are represented. The hindbrain of CTRMO and MCT8MO zebrafish embryos were analyzed for colocalization of *pax6a* and *vegfaa* co-expressing cells (white dotted circles) during BHB development at different time points. Colocalization was determined by using the

colormap colocalization plugin of Fiji software in the region of every CtA. The white arrowhead represents the mid-cerebral vein (MCeV), and the red arrowhead represents the primordial hindbrain channels (PHBC). Numbers 1 – 7 indicate the CtA in its respective rhombomere. Scale bar: 50 μm. During the different time points of hindbrain vasculature development (**c**-CtA1, **d**-CtA2, **e**-CtA3, **f**-CtA4, **g**-CtA5, **h**-CtA6, **i**-CtA7), the presence and absence of CtAs and *pax6a/vegfaa* co-expressing cells were analyzed and the frequency determined. CtAs 2, 4, 5 and 7 correlate with CtA development and *vegfaa/pax6a* co-expressing cells. Fisher's exact test. *n* = 8 (30 hpf CTRMO, 32 hpf MCT8MO), 9 (30 hpf MCT8MO, 36 hpf MCT8MO), 10 (all other stages and conditions). For detailed statistics, see Supplementary Data 2.

and 2, co-expressing *pax6a/thrab*+ cells were found before CtA ingression (Fig. 8h, I, Supplementary Fig 7). These observations shows that MT3 responsivity is not involved in CtA 1 and 2 ingressions (Fig. 7). Still, it cannot be excluded that *thrab* aporeceptor function is not involved in these CtA

ingressions (Fig. 8h, i). In the case of CtA 7, *pax6a/mct8/thraa*+ cells (Figs. 7 and 8, Supplementary Fig 6) are present in rhombomere 7 before ingression, but the vessel ingresses in CTRMO embryos at a time where MT3-sensitive *pax6a*+ cells are not present and independently of *mct8*

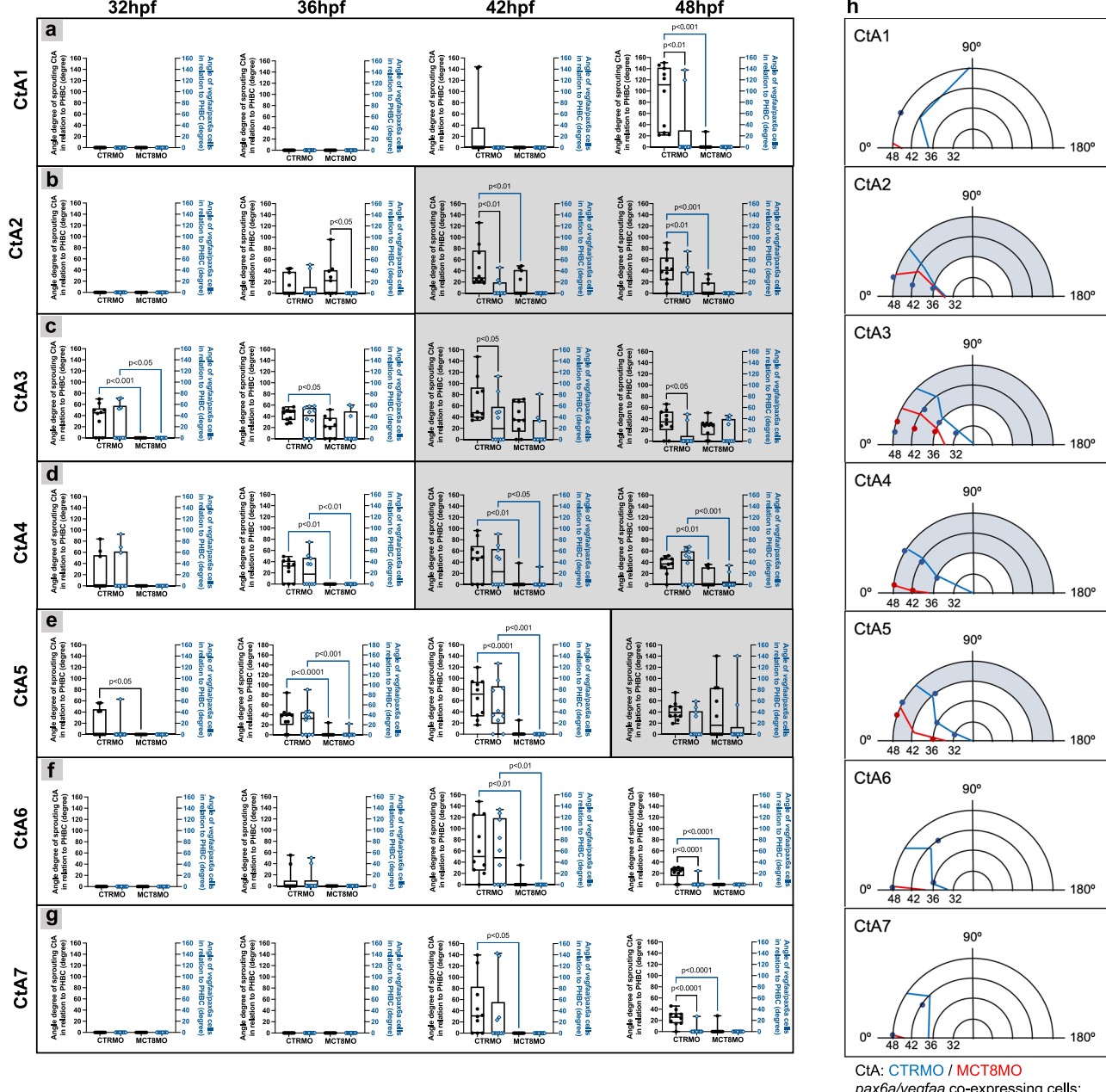

**Fig. 6 | *pax6a/vegfaa* co-expressing cells guide central arteries migration.** The angle of the developing CtA 1 (**a**), CtA 2 (**b**), CtA 3 (**c**), CtA 4 (**d**), CtA 5 (**e**), CtA 6 (**f**), and CtA 7 (**g**) were measured using the PHBC as the basis (0 degrees). The angle was measured before the turnover of the CtA towards the BA or directional change of the CtAs towards its ipsilateral neighbors. CtAs and *pax6a/vegfaa* co-expressing cells that were absent are indicated as 0°. Grey boxes show the CtAs that changed the directional migration during hindbrain vasculature development. **a–g** Statistical significance was determined using 2-way ANOVA. *n* = 8 (30 hpf CTRMO, 32 hpf

MCT8MO), 9 (30 hpf MCT8MO, 36 hpf MCT8MO), 10 (all other stages and conditions). Data are presented as box-and-whisker plot, where the black thick horizontal line represents the median. Each dot represents a biological replicate; error bars represent standard deviations (smallest and highest value). For detailed statistics, see Supplementary Data 2. **h** Graphical representation of the CtA directionality and the position of the *pax6a/vegfaa* co-expressing cells during BHB development. The mean values were used to construct these graphs.

knockdown (Fig. 7). This argues that CtA7 develops independently of MT3 signaling and thyroid receptor aporeceptor function.

CtA 3 presents the most complex situation. The *pax6a/thraa/thrab*+ cells (Fig. 8g, n) are present before and during CtA 3 ingression, irrespective of *pax6a/mct8*+ cells' presence (Fig. 7h). This evidence does not exclude that *thraa* and/or *thrab* or both functions are not involved in CtA 3 ingression. This hypothesis will have to be answered in future studies.

However, for CtAs 4, 5, and 6, there is a temporal pattern of *thraa* and *thrab* expression in *pax6a/mct8*+ cells (Figs. 7e, f and g and 8e, f, g, k, l, and

m). The presence of *pax6a/mct8/thraa/thrab*+ cells coincides with vessel ingression, indicating that MT3 signaling via *thraa* and *thrab* is likely necessary for *pax6a*+ cells attraction of CtAs via *vegfaa* signaling.

**Loss of *pax6a* function leads to impaired hindbrain CtA development**

To confirm the involvement of hindbrain *pax6a* NPCs in hindbrain vasculature development, we knocked out *pax6a*. A guide RNA (gRNA) was designed against the zebrafish *pax6a* exon 7 locus, thus leading to a

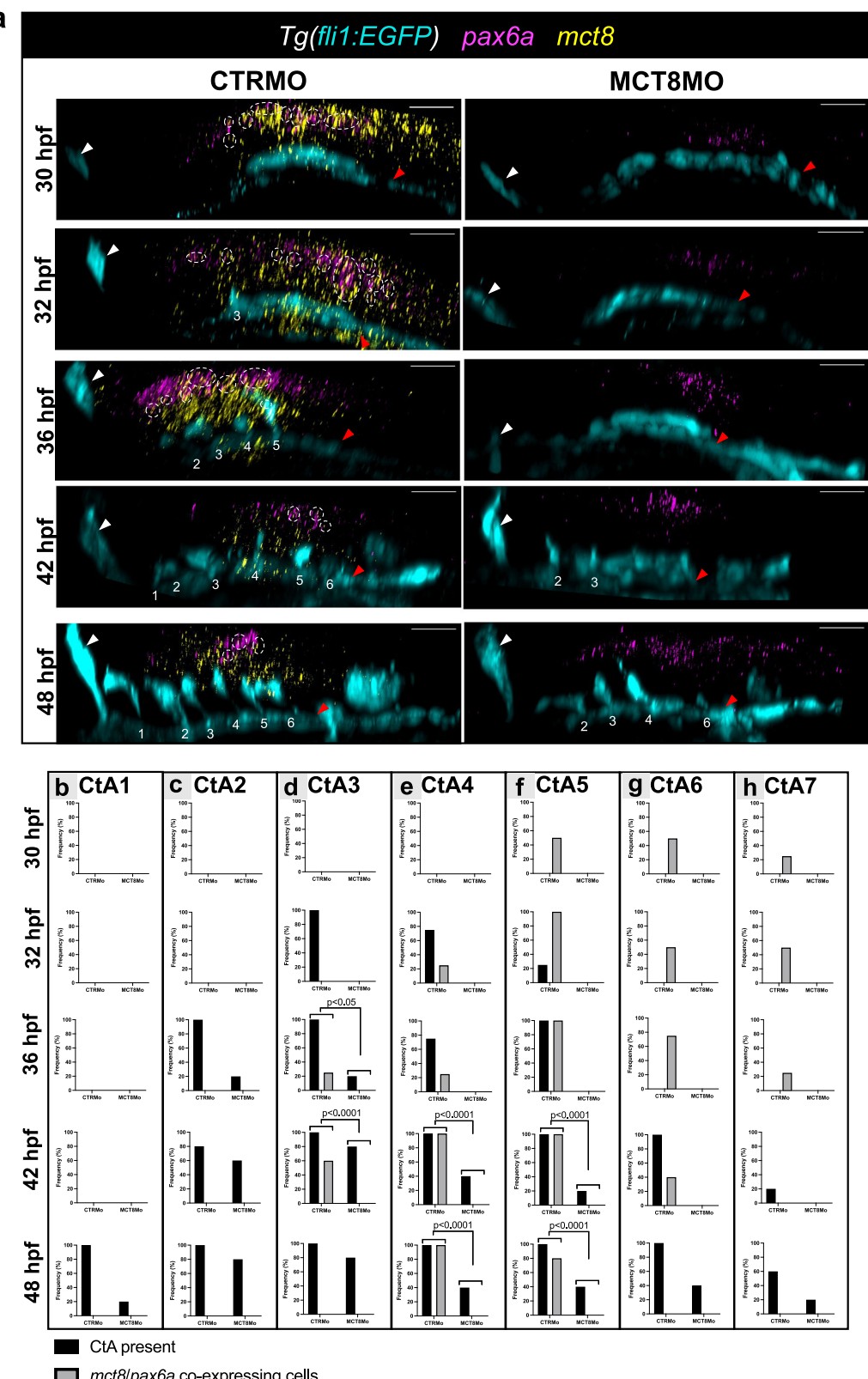

**Fig. 7 | *pax6a*-expressing cells colocalize with T3 transporter *mct8*. a** Lateral view of maximum projection images of fluorescent double WISH against *pax6a* (magenta) and *mct8* (yellow) and immunostaining against GFP (endothelial marker, cyan) in *Tg(fli1:EGFP)* CTRMO and MCT8MO zebrafish embryos at 30, 32, 36, 42 and 48 hpf. The hindbrain of CTRMO and MCT8MO zebrafish embryos were analyzed for colocalization of *pax6a* with *mct8*-expressing cells (white dotted circles) during BHB development at different time points. Colocalization was determined by using the colormap colocalization plugin of Fiji software in the region of every CtA. The white arrowhead represents the mid-cerebral vein (MCeV) and the red arrowhead represents the primordial hindbrain channels (PHBC). Numbers 1 – 7 indicate the CtA in its respective rhombomere. Scale bar: 50 μm. During the different time points of BHB development, the presence and absence of CtA 1 (**b**), CtA 2 (**c**), CtA 3 (**d**), CtA 4 (**e**), CtA 5 (**f**), CtA 6 (**g**), CtA 7 (**h**) and *pax6a/mct8* co-expressing cells were analyzed and the correlation was determined. Statistical significance was determined using Fisher's exact test. n = 3 (36 hpf MCT8MO), 4 (30 hpf CTRMO/ MCT8MO, 32 hpf CTRMO, 36 hpf CTRMO), 5 (32 hpf MCT8MO, 42 hpf CTRMO/ MCT8MO, 48 hpf CTRMO/MCT8MO)). For detailed statistics, see Supplementary Data 2.

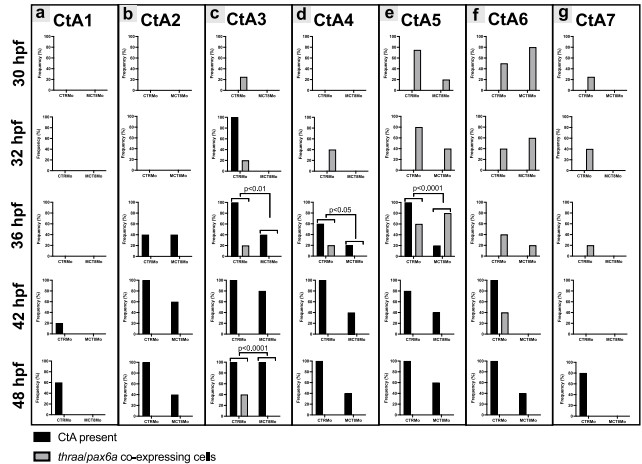

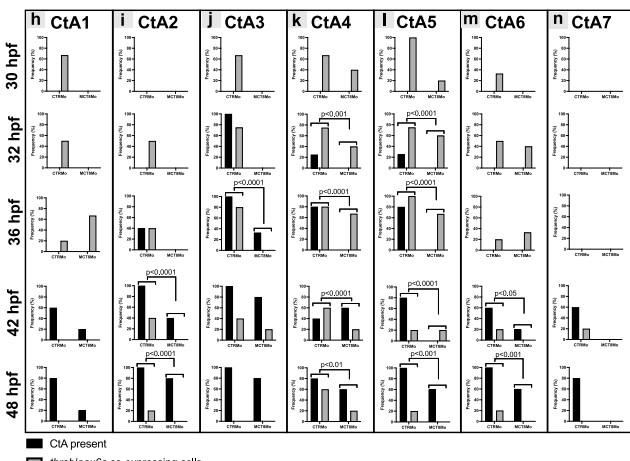

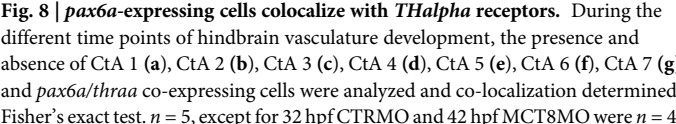

**Fig. 8 | *pax6a*-expressing cells colocalize with *THalpha* receptors.** During the different time points of hindbrain vasculature development, the presence and absence of CtA 1 (**a**), CtA 2 (**b**), CtA 3 (**c**), CtA 4 (**d**), CtA 5 (**e**), CtA 6 (**f**), CtA 7 (**g**) and *pax6a/thraa* co-expressing cells were analyzed and co-localization determined. Fisher's exact test. *n* = 5, except for 32 hpf CTRMO and 42 hpf MCT8MO were *n* = 4.

During the different time points of BHB development, the presence and absence of CtA 1 (**h**), CtA 2 (**i**), CtA 3 (**j**), CtA 4 (**k**), CtA 5 (**l**), CtA 6 (**m**), CtA 7 (**n**) and *pax6a/ thrab* co-expressing cells were analyzed and co-localization determined. Fisher's exact test. *n* = 3 (30 hpf CTRMO, 36 hpf MCT8MO), 4 (32 hpf CTRMO), 5 (all other stages and conditions).

frameshift before the homeobox and the proline/serine/threonine rich (PST) domains that prevents DNA binding[47,48] (Supplementary Fig 8). Zebrafish embryos were injected with the *pax6a* gRNA and Cas9 protein (Weissman Institute, Israel) in 1-cell stage zebrafish embryos. The efficiency of the injection was verified after phenotyping for small eye size, characteristic of impaired *pax6a* function[34], and confirmation by genotyping by PCR (Supplementary Fig 8) of the F0 *pax6a* mutant zebrafish embryos (hereafter called crispants) at 24 hpf.

We performed live imaging of *pax6a* crispants to evaluate CtA development in *Tg(kdrl:CaaX-mCherry)*[23]. We observed that *pax6a* crispants developed a normal PHBC (yellow arrowheads in Fig. 9a lower panel), mid-cerebral vein (MCeV, red arrowheads in Fig. 9a lower panel), and lateral dorsal aorta (LDA, blue arrowhead in Fig. 9a, lower panel). Like MCT8MO *(Tg(fli1:EGFP) background)* zebrafish embryos (Fig. 9b), *pax6a* crispants (Fig. 9a) displayed fewer CtAs than non-injected control embryos. In *pax6a* crispants, the first CtAs sprouts were present at 36 hpf. At 40 hpf, a third CtA developed, and these were the only CtAs that developed until 48 hpf (Fig. 9a, second panel). CtAs in CTRMO zebrafish embryos developed similarly to those in non-injected *Tg(kdrl:CaaX-mCherry)* control embryos (Fig. 9a, b, first panels). MCT8MO zebrafish embryos developed CtAs after 36 hpf, presenting two CtAs at 40 hpf. At 44 hpf, a third CtA developed, and these were the only CtAs that developed until 48 hpf (Fig. 9b second panel). This shows that both *pax6a* crispants and MCT8MO zebrafish embryos presented CtA defects that arise in a similar chronological order (Fig. 9a, b). We further confirmed the results obtained with crispant embryos for *pax6a* loss-of-function in F2 *pax6a* homozygous loss-of-function zebrafish larvae at 4 dpf (Fig. 9c). Further analysis of *pax6a* expression in *mct8* knockout embryos confirms the loss of some but not all hindbrain *pax6a+* cells. Together, this data confirms that *pax6a + NPCs* loss is behind impaired hindbrain vasculature development in both *pax6a+* and *mct8* loss of function models and that MT3 is involved in some but not all *pax6a +* NPCs development and function.

## Discussion

Several studies have shown a relationship between THs and brain vascularization during development. In adult dogs, it has been demonstrated that chronic hypothyroidism leads to increased BBB permeabilization[49,50]. Impaired brain vascularization was also observed in post-natal induced hypothyroid rats[21,51]. In zebrafish, impaired hindbrain vascularization development has been shown in MCT8MO embryos[12] and larvae[14]. Importantly, new evidence from AHDS patients shows a reduced

vascularization of the cortex[16], further confirming that MCT8-dependent T3-signaling is essential for human brain vascularization.

Here we show that hindbrain vasculature development in the zebrafish requires MT3 signaling. Our evidence outlines a mechanism where MT3-dependent development of a specific hindbrain *pax6a + NPCs* population is involved in the *vegfaa*-dependent chemoattraction of endothelial cells ingression into the hindbrain that enables ingression of CtAs (Fig. 10). Our work highlights the integrative role of MT3 in embryonic development, which is necessary for CNS development[12,27,30] but also for integrating the vasculature in the CNS to give rise to a fully functional organism.

Although we did not analyze CtA development after 48 hpf, this phenotype was not a consequence of the late development of the MCT8MO embryos, given that other developmental landmarks, such as heart development, co-occurred with CTRMO zebrafish embryos. Impaired brain vascularization was also observed in MCT8MO zebrafish larvae at 120 hpf[14], confirming that the vascular phenotype persists throughout development. Moreover, our results with the newly generated *mct8* knockout also support the persistent consequences found in the *mct8* knockdown model.

## MT3 regulates hindbrain *vegfaa* expression for CtA development

Vertebrate brain angiogenesis strongly depends on VEGF and their endothelial tyrosine kinase receptors (VEGFR)[38,39,52–54]. In zebrafish two *vegfa* paralogues exist, *vegfaa* and *vegfab*[38]. First, we analyzed the expression of *vegfaa* and *vegfab* by qPCR; however, no significant changes were observed during the BHB developmental time window (28 to 72 hpf) (Supplementary Fig 2c, d). However, WISH expression analysis showed that both genes were reduced in the brain in MCT8MO embryos (Supplementary Fig 3a, b). Nevertheless, in MCT8MO zebrafish embryos, *vegfab* was only mildly affected. At the same time, *vegfaa* expression was significantly reduced in the hindbrain, indicating that MT3 regulates both *vegfa* paralogues but was affecting more the expression of *vegfaa*. Similarly to our results, immunohistochemistry analysis of the ventricular zone of prenatal hypothyroid mice showed significant differences in cerebral microvasculature, COL IV, PECAM-1, and F-actin expression between control and PTU-treated mice[51]. These studies found no differences when bulk transcriptomic analyses were carried out, indicating that T3 action on brain vascularization occurs discretely cellular and context-dependently.

We also showed that *vegfaa-165* mRNA partially rescued the MCT8MO hindbrain vasculature phenotype (Fig. 1e, f). This revealed that MT3 through Mct8 regulates CtA development through *vegfaa* expression. However, the incapacity of *vegfaa-165* mRNA injection to ultimately rescue

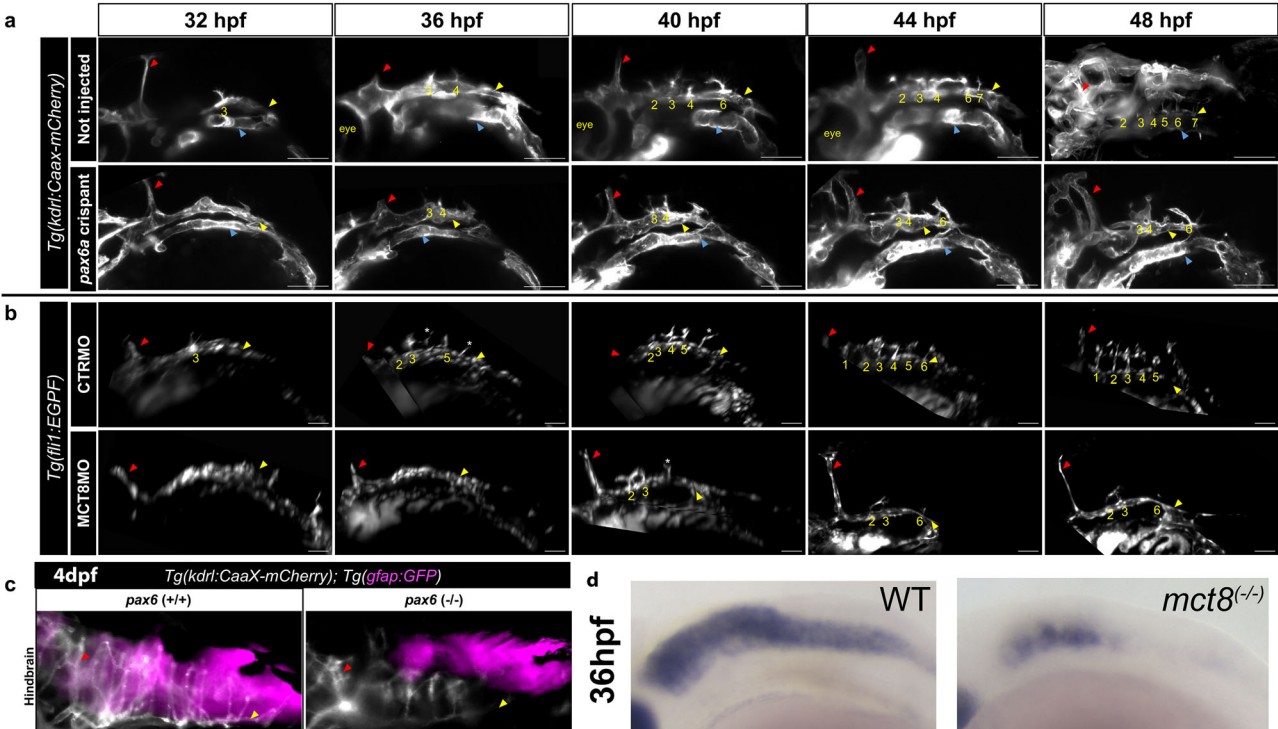

**Fig. 9 | *pax6a* mutant zebrafish embryos show similar hindbrain vascular development defects as MCT8MO zebrafish embryos. a** Live imaging of 32 hpf zebrafish embryo at the start of imaging between non-injected (control) and *pax6a* crispant zebrafish embryos are represented. The vascular system (*kdrl*) is shown in white (mCherry) in the reporter line *Tg(kdrl:CaaX-mCherry)*. Dorsal view of maximum projection images of the mentioned time points is shown. **b** Live imaging of 32 hpf zebrafish embryo at the start of imaging between CTRMO and MCT8MO zebrafish embryos are represented. The vascular system is shown in white (GFP) in the reporter line *Tg(fli1:EGFP)*. Dorsal view of maximum projection images of the mentioned time point is shown. *n* = 2. Scale bar: 50 μm. **c** Comparison between control (not injected) and mutant *pax6a* CRISPR zebrafish larvae at 4 dpf are presented. *n* = 5 (control), 9 (*pax6a* CRISPR). Mutant *pax6a* knockout zebrafish larvae present only 4 CtAs, while the control zebrafish have 7 CtAs. The red arrowhead represents the mid-cerebral vein (MCeV), the yellow arrowhead represents the primordial hindbrain channels (PHBC), and the blue arrowhead represents the lateral dorsal aorta (LDA). White * represents sprouting projections of the PHBC to the BA (due to the inclination of the hindbrain imaging, we can visualize these structures). Numbers 1 – 7 indicate the CtA in its respective rhombomere. **d** In *mct8*[(−/−)] embryos at 36 hpf it is observed a loss of *pax6a* hindbrain cells coincident with CtAs underdevelopment. Scale bar 50 μm.

all CtAs, argues that MT3 might regulate another factor involved in CtA development. Rossi et al. [38] showed that although *vegfaa* is the necessary angiogenic factor in CtA development, *vegfab* knockout embryos also lose some CtAs. The expression of *vegfab* is also affected in *mct8* knockdown embryos, although to a much lesser extent (Supplementary Fig 3b). Another possible factor responsible for the partial rescue of the hindbrain CtAs is the extracellular matrix (ECM) composition. In the 25 hpf MCT8MO zebrafish transcriptome[30], ECM constitutes one of the most affected genetic pathways, arguing for overall differences in ECM composition. Transcriptomic analysis of primary cerebrocortical cells of mice[55] and from microdissected brain tissues of the ventricular zone of hypothyroid rats[51] also identified ECM genes regulated directly by TH. Moreover, the Vegfaa-165 isoform binds to the ECM[56]. Modifying the ECM components might affect the binding capacity of Vegfaa-165 to the ECM[57], leading to a lower concentration gradient or effectivity of *vegfaa* signaling to initiate the angiogenic sprouting of the endothelial cells from the venous PHBC. These results further highlight that the hindbrain vasculature defects observed in MCT8MO zebrafish embryos result from an indirect regulation of MT3 on the endothelial cells. Still, it regulates MT3-responsive hindbrain cells that express *vegfaa*, and likely the extracellular environment that enables adequate Vegfa signaling.

A further example of how indirect regulation of MT3 of the hindbrain vascularization occurs is the recruitment of the pericytes to the different hindbrain vascular structures. Pericytes are essential for blood vessel maturation and BBB stabilization as they express tight junction proteins to promote vascular permeability[58], allowing for the establishment of a competent neurovascular unit. In MCT8MO zebrafish embryos, pericyte numbers were significantly reduced in the PHBC, BA, and CtAs at 48 hpf, and treatment with T3 could not accelerate pericyte migration. These observations are further evidence of an indirect role of MT3 in vascular structures development. MT3 did not directly regulate pericyte recruitment by inducing *pdgfrb* expression, but rather, it affected pericyte recruitment due to impaired hindbrain vasculature development used by these cells to achieve their final position.

### Hindbrain *pax6a* + NPCs are responsible for the chemoattraction of the CtAs in a cell-autonomous way

Maternal THs deficiency leads to an underdeveloped CNS. Several reports have highlighted the importance of maternal THs for NPC expansion, neurogenesis, and the differentiation of neuronal and glial cells, among others[1,46,59]. Supporting the hypothesis that impaired MT3 signaling results in impaired neurodevelopment is the source of chemoattracting CtAs via *vegfaa*; we identify the *vegfaa* signaling source by examining MT3-dependent hindbrain CNS cells. We verified that *pax8* (inhibitory neuron marker) mutant zebrafish embryos presented a complete BHB (Fig. 3d) and that *copine 4* (*cpne4*) interdigit innervating neurons do not express *vegfaa* (Fig. 4f), thus confirming they are not responsible for CtA development. Nonetheless, we identified these neurons (*pax8* and *cpne4*) as MT3-responsive, indicating that their progenitor cells likely depend on MT3 for differentiation.

In mammals, after perineural vascular plexus (PNVP) formation, the sprouting of the blood vessels into the neural tissue is promoted by NPCs that secret Vegf-A[60–62]. In fetal hypothyroid rats, *Pax6* NPCs were reduced[46], like in the hindbrain of MCT8MO zebrafish embryos[30], leading us to

**Fig. 10 | Proposed model for MT3 Mct8-mediated hindbrain development in zebrafish embryogenesis. a** Illustration of the zebrafish hindbrain with proposed action of MT3 signaling through Mct8 on CtA development. The different *pax6a* neural progenitor cells (NPC) identified in the different rhombomeres with the TH machinery *mct8*, *thraa* and *thrab* are illustrated. In rhombomere 1 we suggest the involvement of another central nervous system (CNS) cell type that is MT3 dependent and the source of *vegfaa* required for CtA development. In rhombomere 3, *pax6a* NPCs are present, but MT3 signaling is not involved in CtA 3 development (dashed arrow line) but enhances its development. Vascularization of rhombomeres 4, 5 and 6 suggests that, besides *vegfaa*, another angiogenic factor is involved in the angiogenic sprouting of these CtAs. **b** Illustration of a lateral view of the zebrafish hindbrain showing the effect of Mct8 knockdown and MT3-impaired signaling. The most frequently developed CtAs in the hindbrain of 48 hpf zebrafish embryos are represented. Although significantly reduced in rhombomere 2, *vegfaa* signaling is still present in MCT8MO zebrafish embryos, showing that another CNS cell type, independent on MT3 signaling, is responsible for releasing *vegfaa* or able to compensate for the lack of *pax6a* + /MT3-responsive cells. The inability to rescue the vascularization of rhombomere 4 by *vegfaa-165* mRNA suggests that another angiogenic factor is necessary for its development. C: cerebellum; chb: caudal hindbrain; MHB: Midbrain-hindbrain boundary; MT3: maternal T3; PHBC: Primordial Hindbrain Channels; r1 – r8: rhombomere 1 – 8; sc – spinal cord. The endothelial cell model was adapted from https://bioart.niaid.nih.gov/discover?q=endothelial.

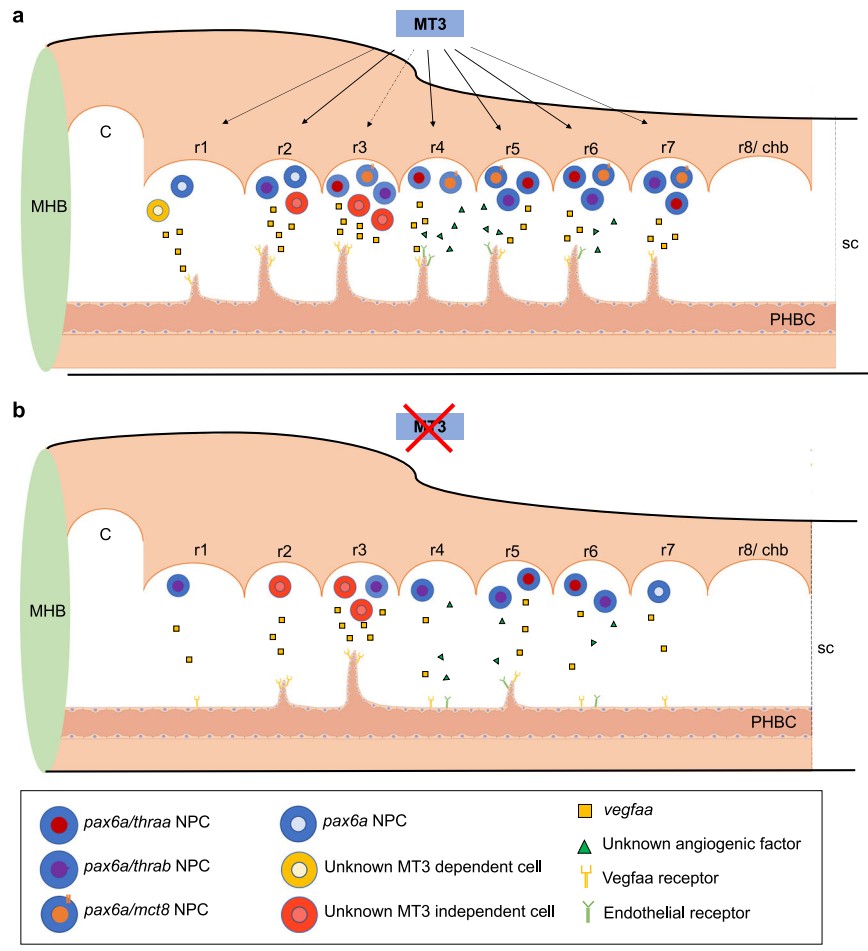

hypothesize that these cells are the source of *vegfaa* and responsible for CtA development in the zebrafish hindbrain. Moreover, in mice, *Pax6* NPCs express thyroid hormone receptor alpha-1 (THRα1)[46], consistent with being under the regulation of MT3. We found a reduced expression of *pax6a* in the hindbrain of MCT8MO zebrafish embryos and also that some of these cells colocalize with *vegfaa*. Our findings show that in contrast to the spinal cord in mice[44] and zebrafish[39], where neurons contribute to vascularization, in the zebrafish hindbrain, this role lies with the NPCs. Our evidence corroborates previous studies showing[63] that zebrafish brain and trunk tip cells express different types of matrix metalloproteinase, leading to different specifications of blood vessel characteristics and developmental programs regulating vascularization.

We identified that CtA migratory routes correlate with *pax6a/vegfaa* co-expressing cells in zebrafish embryos. In MCT8MO embryos, altered routes occurred due to reduced *pax6a/vegfaa* cells, except CtA 3, which maintained its BA connection (Fig. 6). CtA 3's unique persistence in MCT8MO embryos correlated with residual *vegfaa* in rhombomere 3 (Fig. 1a–d) and its essential BA interconnection for hindbrain circulatory closure. CRISPR-generated *pax6a* mutants confirmed CtA 3 development is independent of *pax6a* NPCs (Fig. 9a).

CtA 1 (rhombomere 1) was rescued by *vegfaa-165* mRNA (Fig. 1g) but showed no correlation with *pax6a/vegfaa* cells (Fig. 5c), suggesting another MT3-regulated *vegfaa*-secreting cell type guides its migration into the hindbrain. CtA 2 development ( ~ 50% in MCT8MO) relied on residual *vegfaa* signaling (Fig. 1b-d) and correlated with *pax6a/vegfaa* cells (Fig. 5d), indicating MT3 regulation.

Rhombomere 4 vascularization failed *vegfaa* rescue (Fig. 1g) but correlated with *pax6a/vegfaa* cells (Fig. 5f) and *pax6a/mct8* co-expression

(Fig. 7e), confirming Mct8-mediated MT3 control of CtA 4. Rhombomeres 5 and 6 showed partial *vegfaa* rescue (Fig. 1g) and correlation with *pax6a/vegfaa* cells (Fig. 5g, h) and *pax6a/mct8* for CtA 5 (Fig. 7f), suggesting additional angiogenic factors. CtA 7 rescue via *vegfaa* (Fig. 1g) and *pax6a/vegfaa* correlation confirmed MT3 regulation via *pax6a* NPCs.

*pax6a* loss-of-function experiments validated *mct8* knockdown/knockout results, demonstrating *pax6a* + NPCs direct hindbrain vessel ingression (Fig. 9). These findings reveal cell/tissue-specific roles for MT3 signalling.

Interestingly, *mct8*, *thraa*, and *thrab* were expressed in a spatio-temporal manner, indicating that these cells are responding autonomously to MT3. For CtAs 4, 5, and 6, vessel ingression occurred only when *pax6a* + NPC co-expressed the three T3-signaling genes. This evidence supports that a change from thyroid receptor apo- to receptor function is involved in coordinating the sprouting and ingression of the CtAs for each rhombomere in a timely coordinated manner. Nonetheless, this new evidence needs to be functionally validated in future studies. This shows that, as has been described[37], while CtAs sprouting is variable, it is not random.

Our analysis shows that an MT3-dependent function on *pax6a* + NPCs is required for CtA 4, 5, and 6 ingressions. For the remaining CtA, the situation is more complex. Except for CtA 1 and 7, independent of MT3 action, there is a correlation between vessel ingression and *pax6a* + NPCs co-expressing *thrab*. Nonetheless, *thrab* is considered to have a dominant negative function[64], thus making a definitive conclusion about the role of MT3 in these NPCs challenging to determine from our data. These need additional studies in the future. This study indicates that not all MT3-dependent NPCs can mediate brain vascularization after MT3-signaling. Notably, in our dataset, only *mct8/thraa/thrab*+ *pax6a* + NPCs seem to be

involved in hindbrain vascularization, suggesting that a particular thyroid receptor-mediated MT3-action is crucial for giving *pax6a* + NPCs the functional capacity to attract blood vessels.

## Relevance to AHDS

T3-signaling genes are not expressed in the developing zebrafish vasculature until 48 hpf [12], but impaired MT3-signaling in the CNS affects hindbrain vasculature development and angiogenic-related developmental pathways. Similarly, human endothelial cells derived from AHDS patient iPSC do not present altered differentiation [15], and recent evidence from AHDS patient-derived organoids shows that the neurological effects of impaired *MCT8* function are the result of MT3 action on neural cells and not impaired BBB function [13]. However, and at least in humans, it cannot be discarded that T4 might have a role in AHDS. Our work demonstrates that MT3 regulates a particular population of *pax6a* + NPCs, defined by co-expression of *mct8/thraa/thrab*, that results in *vegfaa* expression and chemoattraction and ingression of endothelial cells from the PHBC, allowing for the sprouting of CtAs 4, 5 and 6 (Fig. 10). We did not analyze whether MT3 directly affects endothelial cell proliferation during this study. However, this seems unlikely considering previous results showing that no TH-signaling genes are expressed in these cells at the zebrafish developmental stages [12]. Nonetheless, our results indicate that there are indirect effects. Specifically, our findings show a decrease in *vegfaa*-mediated attraction in the endothelial sprouting and migration mechanism.

The present study, together with evidence from AHDS patients [16], argues that the pathophysiological consequences of MT3-impaired signaling are not only related to decreased neural cell diversity [13,27,30], but also to the consequences on CNS vascularization. Given the irreversible neurological damage found in AHDS patients that cannot be recovered after birth [1,65] it is very likely that the consequences on brain vascularization also cannot be recovered in these patients. Moreover, given the widespread prevalence of thyroid deficiencies that constitute the most common cause of preventable impaired cognitive development [66], it cannot be excluded that some of the impacts in affected individuals also comprise impaired brain vascular development.

The fact that the BHB is disrupted in zebrafish argues that brain permeabilization might be compromised, as has been observed in hypothyroid dogs [49,50], leading to the access of several factors/hormones/wastes to the brain. Nonetheless, the human AHDS brain maintains a hypothyroid state while the peripheral tissues are hyperthyroid [67], suggesting impaired BBB development maintains some barrier function. This seems to be the case in zebrafish as well. We never observed any brain edema in *mct8* morphant zebrafish embryos [12]. Although pericyte recruitment is delayed into the BHB, it still occurs, further arguing that in zebrafish, the BHB permeability is maintained, at least at the end of embryogenesis.

## Conclusion

The presence of MT3 allows the survival and proliferation of specific *pax6a* NPCs in the zebrafish hindbrain, which are necessary for the timely ingression of the CtAs through the expression of the angiogenic factor *vegfaa*. The presence of the different TH signaling genes (*mct8*, *thraa* and *thrab*) in *pax6a* NPCs throughout development and specific rhombomeres indicates that MT3 acts upstream of CtA ingression, contributing to a fully functional hindbrain vasculature development (Fig. 10).

## Limitations of this study

Although we have uncovered the developmental mechanism by which MT3 controls the development of the hindbrain vasculature and identified the cells under the hormone's control, we cannot determine the exact identity of the *pax6a* + NPC that responds to and mediates MT3's action on CtA ingression. Moreover, we cannot determine the genetic and developmental causes that lead to the loss of the specific *pax6a* + NPC under the regulation (or lack) of MT3 signaling. This is the subject of ongoing research.

## Data availability

All source data used to produce the graphs in this work is provided in Supplementary Data 1. Supplementary tables and figures can be found in Supplementary information. The statistical data can be found in Supplementary Data 2.

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

## Acknowledgements
This study received Portuguese national funds from FCT - Foundation for Science and Technology through project PTDC/EXPL/MAR-BIO/0430/2013. ABC-RI CRESC Algarve 2020 and internal ABC financing. MT was a recipient of the FCT PhD grant SFRH/BD/108842/2015, NS was a recipient of the FCT PhD grant SFRH/BD/111226/2015. MAC received an FCT-IF Starting Grant (IF/01274/2014) and 2016 Collaborative Research (A1) from National Institute of Genetics (Japan). Support of ABC and Camara Municipal de Loulé. We thank NBRP from the Ministry of Education, Culture, Sports, Science and Technology of Japan, and JSPS KAKENHI JP24K02008 (to KK). We thank Brant Weinstein, Ching-Ling Lien, Wolfgang Driever, Wiebke Herzog, Raquel Andrade, and Sachiko Takayama for plasmids. We thank Julien Vermot, António Jacinto and Fumihito Ono for zebrafish lines.

## Author contributions
Marlene Trindade: Methodology, Visualization, Writing - Original Draft, Review & Editing. Nádia Silva: Methodology, Visualization, Review & Editing; Joana Rodrigues: Methodology, Visualization, Review & Editing; Koichi Kawakami: Methodology, Financing, Review & Editing; Marco A. Campinho: Conceptualization, Financing, Methodology, Writing - Original Draft, Review & Editing, Supervision. All authors read and approved the final manuscript.

## Competing interests
The authors declare no competing interests.
