## [Transparent Peer Review file · Communications Biology]

Maternal thyroid hormone is required to develop the hindbrain vasculature in zebrafish.

Corresponding Author: Dr Marco António Campinho

Version 0:

Reviewer comments:

Reviewer #1

(Remarks to the Author)

Review Comments

Previous studies showed that TH is essential for vascularizing several organs via VEGFA regulation, while during neurodevelopment which cells mediate vascularization are mostly unknown. Here Trindade et al proved that hindbrain *pax6a+* neuroprogenitors cells (NPCs), but not neurons expressing *vegfaa* to instruct central arteries (CtAs) ingression into the hindbrain. This discovery was not reported before. The manuscript presented a well-designed study with robust data that not only advances the understanding of hindbrain vascular development but also provides insights into potential mechanisms underlying human diseases, such as Allan-Herndon-Dudley Syndrome (AHDS). Although my own main work is not in the filed of vascular development and I could not find some critical issues, I think this work is very important. Below are some issues should be addressed.

- 1, the manuscript should be numbered for each line, which would be benefit for review process. It is hard to review the current manuscript.
- 2, the current data could not support the claim that "Maternal" thyroid hormone is required to develop the hindbrain vasculature (In the tile and another section et al). We suggested to delete the word "maternal", or use the other proper presentation.
- 3.The results section (e.g., pages 4–7) is overly detailed, with redundant descriptions and an excessive number of figures. For instance, the description of *mct8* knockout effects on hindbrain vasculature is repetitive, and some figures (e.g., Figures 1 and 2) could be consolidated. Simplify the textual descriptions to highlight key findings, such as: "*mct8* deficiency reduces CtA numbers and *vegfaa* expression, demonstrating MT3's pivotal role in hindbrain vascular development." Consider merging sub-panels from Figures 1 and 2 to reduce redundancy.
- 4.In Fig. 1e, whether the vascular defects caused by *mct8* loss are related to endothelial cell functon(e.g. proliferation or migration)?
- 5.Many acronyms appear in the text without explanation, such as BBB, BA, CTRMO, MCT8MO.
- 6.Should we consider testing if the thyroid hormone was changed in MCT8MO and *mct8* mutants embryos?
- 7.The discussion section lacks in-depth analysis of the findings. Specifically, the dynamic role of *pax6a+/vegfaa* cells in CtA development and its potential relevance to mammalian models is insufficiently addressed.
- 8.The manuscript does not include an ethical statement for the zebrafish experiments. We suggested to Include a clear statement of ethical approval for animal experiments, specifying the institution and guidelines followed.
- 9.The current data sufficiently support the main conclusions. If possible we suggested the following supplementary experiments could significantly enhance the manuscript:
Functional validation: Perform cell-specific knockdown or rescue experiments (e.g., CRISPR or chemical inhibitors) to directly validate the role of *pax6a+/vegfaa* cells in CtA development.
- 10.I am not a native speaker, while I still think there are some language problems. Some parts in the manuscript are not easy to understand. Some sentences are verbose or repetitive, impacting readability. For example, on page 5: "Expression of *vegfaa* during the different developmental stages where CtA ingression (32, 36, and 48 hpf) was reduced in MCT8MO embryos..." To this sentence we suggested to revise it as : "Vegfaa expression was reduced at 32, 36, and 48 hpf in MCT8MO embryos, particularly in CtA sprouting regions."

This manuscript addresses an important research question and provides robust data to support its conclusions. However, improvements in data presentation, discussion depth and language clarity are needed.

Reviewer #3

(Remarks to the Author)

Trinidade and colleagues have submitted an elegant and robust study, providing evidence on MT3 indirect regulation of zebrafish hindbrain vasculature development and identifying the population that carries out that regulation by vegfaa signaling. It sheds light on TH regulation of angiogenesis and opens new lines of research regarding that subject. Although the results are clear and well-connected, there are some issues in the text that need to be corrected.

General comments:

1. Please revise all abbreviations. I assume a previous version of the paper had M&M before the results section. But now, a lot of abbreviations are not defined as they first appear in the text.
2. Please revise gene and protein nomenclature. Particularly, but not only, when referring to human genes and proteins, there are several examples where caps are absent and cursives too.
3. I suppose that it is based on the clarity of the image, but for me, the fact that the fli fluorescence is depicted in 3 different colors throughout the manuscript, makes it more difficult to follow. Would it be possible to change the figures to make the colors consistent?

Abstract:

1. AHDS symptomatology is not caused only by a lack of maternal thyroid hormone signaling. This needs to be rephrased, as a lot of AHDS alterations arise postnatally due to the lack of proper postnatal MCT8 transport.
2. MCT8 is normally referred to as monocarboxylate transporter 8 or monocarboxylic acid transporter 8, please correct to one of them.
3. Impaired brain development is not the only physiopathological basis of AHDS as it overlooks for example peripheral symptomatology, "one of the hallmarks" or "one of the main physiopathological alterations" may be more correct.
4. In humans MCT8 also transports T4.
- 5 Neuroprogenitor cells (population?). Also, NPC, neuro progenitor, neuroprogenitor cells or neuroprogenitors are used several times in the whole manuscript, please be consistent in using one over the others (maybe NPC as it was defined already).

Introduction:

As there are several comments and line numbers are absent I will follow the text, so a later comment means a later point in the text.

1. Maternal thyroid Hormone(s)
2. Although MCT8 function and its link to the origin of AHDS are hinted in the abstract, the first paragraph of the introduction should include a phrase or two explaining this before the "AHDS patients present..." phrase
3. Reference 7 does not belong here, it does not discuss disease severity in patients.
4. One of the most prevalent would be more accurate (see reference 2).
5. I find Vancamp et al 2020 a more suitable reference for this myelin section as it nicely reviews MRI phenotype in the disease (PMID: 32477268). moreover, MRI is the only method to look at myelin alterations in patients, I would suggest acknowledging this and including either Valcarcel-Hernandez et al. PMID: 34838669 or Lopez-Espindola et al. PMID: 25222753.
6. Again, as in the abstract, MT3 transport is not the only molecular event behind AHDS, please rephrase.
7. I find that affirmations regarding reference 11 are too strong. The T3-independent differentiation is more a suggestion than a real statement, as other TH signaling tools are available for these cells.
8. Thyroid hormones (THs) was not defined that way before.
- 9 Heart should go after mice in the sentence after reference 13.
10. Leads to 'increased' and not decreased VEGFA expression.
11. Please rephrase the sentence before "In this work..."

Results:

MT3 action during zebrafish hindbrain vasculature development

1. Previous, not previously
2. After "indicating that potential MT3 action on CtAs is specific to the brain" I miss a reference, either to a figure or bibliographical.
3. Missing reference for the generation of the Tg Fli line.
4. Ingress only with an r.
5. "but significantly reduced" should be accompanied by Figure 1a-d instead of a only.
6. Please define frequency in Figure 1
7. "vegfaa is likely not the only signal" has an extra dot afterwards.
8. "We quantified the number of pericytes "following" WISH" may be better to avoid "after" twice.
9. The authors assume that pericyte migration is not due to effects on these cells in particular. Could you add a bit more of justification for this statement?
10. The claim of pericyte migration being due to delayed vascular development and not thyroid hormone seems too strong for me. Thyroid hormone metabolism is not addressed in these cells, and as they have been demonstrated to express MCT8, they could very well be expressing Dio3, to avoid T3 overloads in a particularly sensitive tissue, such as what happens in neurogenic niches in mammals. This matter should be rephrased here or at least properly discussed in the discussion.
11. Normalize figure nomenclature with or without caps.
12. We 'wondered' if these cells?
13. I find Supp. Figure 5 of interest to be considered as main figure.

14. Figure 5. Red instead of yellow arrowhead?

Discussion:

1. I would prefer "TH-signaling is essential for human vascularization". (As MCT8 transports both T3 and T4 and the study does not discard T4 actions)
2. "also support the persistent consequences of mct8 knockdown further confirming the observations made with the MCT8 knockdown model" seems a bit redundant, please rephrase.

Version 1:

Reviewer comments:

Reviewer #1

(Remarks to the Author)

I have no comments now.

Reviewer #3

(Remarks to the Author)

First, I want to thank the authors for thoroughly revising the multiple points from my previous review of the manuscript. For me there is only one issue left that I would like to address, not only as a suggestion, but also as a discussion topic of value for the MCT8 community.

Aside from this, I find the manuscript form has been improved, and that this well-rounded study will be of interest to the field.

Abstract:

4. In humans MCT8 also transports T4.

In zebrafish it has been clearly shown after in vitro assays that at zebrafish physiological temperature (28.5C) zebrafish MCT8 can only transport T3 (PMID: 21952246). The biogenesis of thyroid hormones only starts at ~60hpf (PMID: 23022354). Furthermore, zebrafish eggs have available thyroid hormones before fertilization (PMID: 22379985). Collectively, this data proves that thyroid hormones, and namely T3, available for the embryo up until 48hpf (the oldest timepoint analyzed in our study) only has maternal origin. Although in vitro assays with human MCT8 have been shown to be able to transport both T4 and T3, a recent study using brain organoids derived from AHDS patients demonstrates that mutated human MCT8 specifically impairs T3 transport in those cells, further arguing that impaired T3 transport is the physiological problem that gives rise to AHDS (PMID: 38376950). We further strengthen these issues in the revised manuscript

First of all, thank you for the changes in the abstract regarding this issue, it fits better to me. I just wanted to discuss this claim a bit. They do not rule out T4 alterations. They prove T3, that is sure. However, when using DIO2 as a proxy to T4, they acknowledge the fact that they have a problem with CRYM sequestering T4 into the cells, and although they prove that point, they do not pursue the experiment with/without MCT8, so they show no real information about T4 transport or action. This impedes ruling out T4 as a factor.

Introduction:

6. Again, as in the abstract, T3 transport is not the only molecular event behind AHDS, please rephrase.

We have addressed this issue in the revised manuscript. Please also refer to our comment on Abstract point 4.

So continuing with my previous reply, even though I find the phrase better this way, I still find the T3-only explanation rather incomplete, and even the authors of the cited reference seem to think so, as their statement is much milder, giving it as a potential key factor:

"In conclusion, here we provided definitive evidence that MCT8 is critical for early neurogenesis, mediating the bulk of the T3 uptake into developing human neural cells. Such perturbations and altered mechanisms likely play a key role in the pathogenesis of AHDS."

Discussion:

1. I would prefer "TH-signaling is essential for human vascularization". (As MCT8 transports both T3 and T4 and the study does not discard T4 actions).

We understand the reviewers' concerns. But again, I recall our previous answer to point 4 of the abstract. Moreover, in zebrafish embryos and brain organoids of AHDS patients, T3 impaired transport rather than T4 seems to be the pathophysiological origin of this condition, at least at the neurological level, which is what we are focusing on. This does not mean that in other cellular/tissue contexts, T4 transport is not impaired. However, this was not the aim of the present work. Referring to my previous reply, Salas-Lucia and colleagues do not rule out a T4 factor. However, I would like to add some more data to back my disagreement. In reference 9, you can find fetal brain TH levels in an MCT8 deficient patient postmortem sample (which also adds some human in vivo context): "Tissue-specific alteration of iodothyronines and their deiodinases. T4, T3, and rT3 concentrations were reduced by 50% in the cerebral cortex of the MCT8-deficient fetus, but T3 and T4 were normal in the liver. The hormonal deficiency in brain produced the expected increase in type 2 deiodinase and decrease in type 3 deiodinase mRNA expression (Supplemental Table 1)."

Thus, in this same cellular-tissular context, you have altered T4 and DIO2. I do not deny that T3 is a key factor, even the most important, but T4 cannot be ruled out as a potential factor contributing to AHDS.

Reviewers' comments:

Reviewer #1 (Remarks to the Author):

Review Comments

Previous studies showed that TH is essential for vascularizing several organs via VEGFA regulation, while during neurodevelopment which cells mediate vascularization are mostly unknown. Here Trindade et al proved that hindbrain pax6a+ neuroprogenitors cells (NPCs), but not neurons expressing vegfaa to instruct central arteries (CtAs) ingress into the hindbrain. This discovery was not reported before. The manuscript presented a well-designed study with robust data that not only advances the understanding of hindbrain vascular development but also provides insights into potential mechanisms underlying human diseases, such as Allan-Herndon-Dudley Syndrome (AHDS). Although my own main work is not in the filed of vascular development and I could not find some critical issues, I think this work is very important. Below are some issues should be addressed.

First and foremost we would like to thank the reviewer for the insightfull and constructive comments on the manuscript. Bellow we address each comment in detail.

1, the manuscript should be numbered for each line, which would be benefit for review process. It is hard to review the current manuscript.

We have addressed this issue in the revised manuscript.

2, the current data could not support the claim that “Maternal” thyroid hormone is required to develop the hindbrain vasculature (In the tile and another section et al). We suggested to delete the word “maternal”, or use the other proper presentation.

We respectfully disagree with the reviewer. It has been clearly shown after in vitro assays that at zebrafish physiological temperature (28.5C) zebrafish mct8 can only transport T3 (PMID: 21952246). The biogenesis of thyroid hormones only starts at ~60 hpf (PMID: 23022354). Furthermore, zebrafish eggs have available thyroid hormones before fertilization (PMID: 22379985). Collectively, this data proves that thyroid hormones, and namely T3, available for the embryo up until 48 hpf (the oldest time-point analyzed in our study) only have a maternal origin. Moreover, a recent study using brain organoids derived from AHDS patients demonstrates that mutated human MCT8 specifically impairs T3 transport in those cells, further arguing that impaired T3 transport

is the physiological problem that gives rise to AHDS (PMID: 38376950). Therefore, we would stand with our claim in the tile of the manuscript.

3.The results section (e.g., pages 4–7) is overly detailed, with redundant descriptions and an excessive number of figures. For instance, the description of mct8 knockout effects on hindbrain vasculature is repetitive, and some figures (e.g., Figures 1 and 2) could be consolidated. Simplify the textual descriptions to highlight key findings, such as: "mct8 deficiency reduces CtA numbers and vegfaa expression, demonstrating MT3's pivotal role in hindbrain vascular development." Consider merging sub-panels from Figures 1 and 2 to reduce redundancy.

We understand the reviewers' point. However, we believe that combining the two figures will reduce the size of the images and graphs to such an extent that it would be very difficult for readers to discern the details of the observations. In this format, we believe that the readability of the manuscript and the data is enhanced, and we would prefer to maintain it as is.

4.In Fig. 1e, whether the vascular defects caused by mct8 loss are related to endothelial cell functon (e.g. proliferation or migration)?

We find the reviewer's question very pertinent. We do not rule out that this could be the case. However, if this occurs, it is not directly dependent on T3 given that blood-vessels do not express any thyroid signaling gene during embryogenesis (up until 48 hpf). We have in the past demonstrated this and showed that only at 48 hpf some endothelial cells in the hindbrain vasculature start to express mct8 but no thyroid receptor (PMID: 24877564) arguing that the action of T3 on vascular development, at least in the hindbrain, is indirect. I believe that the evidence in the manuscript substantially supports this. We are not sure whether there is a direct influence on endothelial proliferation. Nonetheless, we have never seen changes in endothelial cell proliferation on our datasets (unpublished data). Our evidence suggests exactly this regarding migration. Nevertheless, we have not seen any evidence that this occurs by modulating autonomous endothelial cell migration mechanisms but instead by chemoattraction by other cells and where VEGF signaling is affected. This is one of the findings in this manuscript. We have now addressed this issue in the discussion.

5.Many acronyms appear in the text without explanation, such as BBB, BA, CTRMO, MCT8MO.

We apologise for this oversight. This has been corrected.

6. Should we consider testing if the thyroid hormone was changed in MCT8MO and mct8 mutants embryos?

That would be desirable. We are now establishing a methodology to do this in the future. The reason is that in the absence of transport the hormone cannot get into target cells, leaving it in the yolk, and not used and could indicate that levels will not change. On the other hand, we cannot rule out that, T3 could diminish in both given that dio3 are active during embryogenesis (PMID: 24467742). However, we are not ready to do thyroid hormone quantifications at this time.

7. The discussion section lacks in-depth analysis of the findings. Specifically, the dynamic role of pax6a+/vegfaa cells in CtA development and its potential relevance to mammalian models is insufficiently addressed.

We have addressed this issue in the discussion.

8. The manuscript does not include an ethical statement for the zebrafish experiments. We suggested to include a clear statement of ethical approval for animal experiments, specifying the institution and guidelines followed.

We provided this information in the “Materials and Method section,” sub-section “Zebrafish husbandry and spawning” and added the ethical approval in the revised manuscript: “All experiments and fish husbandry were conducted in accordance with the EU Directive 2010/63/EU on protecting animals used for scientific purposes.” Given that EU and Portuguese regulations indicate that work on zebrafish embryos and larvae up to 5 days post-fertilization does not require submission to the local ethical committee, we have met all legal requirements. Furthermore, all authors who have handled live embryos are duly trained and possess permits for animal experimentation.

9. The current data sufficiently support the main conclusions. If possible we suggested the following supplementary experiments could significantly enhance the manuscript: Functional validation: Perform cell-specific knockdown or rescue experiments (e.g., CRISPR or chemical inhibitors) to directly validate the role of pax6a+/vegfaa cells in CtA development.

We thank the reviewer for these pertinent questions. We have partially done what the reviewer is suggesting. We carried out rescue experiments with vegfaa (Fig. 1E-G) and we demonstrated after pax6a knockout that pax6a+ NPC are the cells under the regulation of maternal T3 that give rise to chemoattraction of central arteries into the hindbrain. We agree that carrying out a cell specific knockout of pax6a+ NPC that are responding to

maternal T3 is the best case scenario. However, we still do not have an unequivocal identification of the identity of maternal T3 responding to pax6a+ NPC. We are in an effort to respond to that with single-cell omics. Regarding chemical inhibition, to my knowledge, there is no any mct8-specific chemical available. Therefore, we consider that using non-specific monocarboxylic acid transporters chemical inhibitor would make the interpretation of the results far more difficult and would not help into dissecting the specific action of maternal T3 and the way MCT8 is involved in this response.

10. I am not a native speaker, while I still think there are some language problems. Some parts in the manuscript are not easy to understand. Some sentences are verbose or repetitive, impacting readability. For example, on page 5: "Expression of vegfaa during the different developmental stages where CtA ingression (32, 36, and 48 hpf) was reduced in MCT8MO embryos..." To this sentence we suggested to revise it as: "Vegfaa expression was reduced at 32, 36, and 48 hpf in MCT8MO embryos, particularly in CtA sprouting regions."

We thank the reviewer for this comment and we have addressed this issue in the revised manuscript.

This manuscript addresses an important research question and provides robust data to support its conclusions. However, improvements in data presentation, discussion depth and language clarity are needed.

We would like to thank the reviewer for the insightful comments and its appreciation of the importance of this work.

Reviewer #3 (Remarks to the Author):

Trindade and colleagues have submitted an elegant and robust study, providing evidence on MT3 indirect regulation of zebrafish hindbrain vasculature development and identifying the population that carries out that regulation by vegfaa signaling. It sheds light on TH regulation of angiogenesis and opens new lines of research regarding that subject.

Although the results are clear and well-connected, there are some issues in the text that need to be corrected.

We would like to thank the reviewer for their insightful and constructive comments on our work. We particularly appreciate the recognition of the scientific importance of our findings.

General comments:

1. Please revise all abbreviations. I assume a previous version of the paper had M&M before the results section. But now, a lot of abbreviations are not defined as they first appear in the text.

We have addressed this issue in the revised manuscript.

2. Please revise gene and protein nomenclature. Particularly, but not only, when referring to human genes and proteins, there are several examples where caps are absent and cursives too.

We have addressed this issue in the revised manuscript.

3. I suppose that it is based on the clarity of the image, but for me, the fact that the fluorescence is depicted in 3 different colors throughout the manuscript, makes it more difficult to follow. Would it be possible to change the figures to make the colors consistent?

We have addressed this issue in the revised figures.

Abstract:

1. AHDS symptomatology is not caused only by a lack of maternal thyroid hormone signaling. This needs to be rephrased, as a lot of AHDS alterations arise postnatally due to the lack of proper postnatal MCT8 transport.

We have addressed this issue in the revised manuscript.

2. MCT8 is normally referred to as monocarboxylate transporter 8 or monocarboxylic acid transporter 8, please correct to one of them.

We have addressed this issue in the revised manuscript.

3. Impaired brain development is not the only physiopathological basis of AHDS as it overlooks for example peripheral symptomatology, "one of the hallmarks" or "one of the main physiopathological alterations" may be more correct.

We have addressed this issue in the revised manuscript.

4. In humans MCT8 also transports T4.

In zebrafish it has been clearly shown after in vitro assays that at zebrafish physiological temperature (28.5C) zebrafish MCT8 can only transport T3 (PMID: 21952246). The biogenesis of thyroid hormones only starts at ~60hpf (PMID: 23022354). Furthermore, zebrafish eggs have available thyroid hormones before fertilization (PMID: 22379985). Collectively, this data proves that thyroid hormones, and namely T3, available for the embryo up until 48hpf (the oldest timepoint analyzed in our study) only has maternal origin. Although in vitro assays with human MCT8 have been shown to be able to transport both T4 and T3, a recent study using brain organoids derived from AHDS patients demonstrates that mutated human MCT8 specifically impairs T3 transport in those cells, further arguing that impaired T3 transport is the physiological problem that gives rise to AHDS (PMID: 38376950). We further strengthen these issues in the revised manuscript.

5 Neuroprogenitor cells (population?). Also, NPC, neuro progenitor, neuroprogenitor cells or neuroprogenitors are used several times in the whole manuscript, please be consistent in using one over the others (maybe NPC as it was defined already).

We have addressed this issue in the revised manuscript.

Introduction:

As there are several comments and line numbers are absent I will follow the text, so a later comment means a later point in the text.

1. Maternal thyroid Hormone(s)

We have addressed this issue in the revised manuscript.

2. Although MCT8 function and its link to the origin of AHDS are hinted in the abstract, the first paragraph of the introduction should include a phrase or two explaining this before the "AHDS patients present..." phrase

We have addressed this issue in the revised manuscript.

3. Reference 7 does not belong here, it does not discuss disease severity in patients.

We have addressed this issue in the revised manuscript.

4. One of the most prevalent would be more accurate (see reference 2).

We have addressed this issue in the revised manuscript.

5. I find Vancamp et al 2020 a more suitable reference for this myelin section as it nicely reviews MRI phenotype in the disease (PMID: 32477268). moreover, MRI is the only method to look at myelin alterations in patients, I would suggest acknowledging this and including either Valcarcel-Hernandez et al. PMID: 34838669 or Lopez-Espindola et al. PMID: 25222753.

We have addressed this issue in the revised manuscript.

6. Again, as in the abstract, MT3 transport is not the only molecular event behind AHDS, please rephrase.

We have addressed this issue in the revised manuscript. Please also refer to our comment on Abstract point 4.

7. I find that affirmations regarding reference 11 are too strong. The T3-independent differentiation is more a suggestion than a real statement, as other TH signaling tools are available for these cells.

We have address this concern. Nonetheless, in page 835 of reference 11 it is stated: “*transcriptomic analyses suggested a high similarity in control and MCT8-deficient iBMECs*”.... “*and they suggest that MCT8 and/or TH are not necessary for the induction and maturation of BBB properties during differentiation*”.

8. Thyroid hormones (THs) was not defined that way before.

We have addressed this issue in the revised manuscript.

9 Heart should go after mice in the sentence after reference 13.

We have addressed this issue in the revised manuscript.

10. Leads to 'Increased' and not decreased VEGFA expression.

We have addressed this issue in the revised manuscript.

11. Please rephrase the sentence before "In this work..."

We have addressed this issue in the revised manuscript.

Results:

MT3 action during zebrafish hindbrain vasculature development

1. Previous, not previously

We have addressed this issue in the revised manuscript.

2. After "indicating that potential MT3 action on CtAs is specific to the brain" I miss a reference, either to a figure or bibliographical.

We have addressed this issue in the revised manuscript.

3. Missing reference for the generation of the Tg Fli line.

We have addressed this issue in the revised manuscript.

4. Ingress only with an r.

We have addressed this issue in the revised manuscript.

5. "but significantly reduced" should be accompanied by Figure 1a-d instead of a only.

We have addressed this issue in the revised manuscript.

6. Please define frequency in Figure 1

We have addressed this issue in the revised manuscript.

7. "vegfaa is likely not the only signal" has an extra dot afterwards.

We have addressed this issue in the revised manuscript.

8. "We quantified the number of pericytes "following" WISH" may be better to avoid "after" twice.

We have addressed this issue in the revised manuscript.

9. The authors assume that pericyte migration is not due to effects on these cells in particular. Could you add a bit more of justification for this statement?

We have addressed this issue in the revised manuscript.

10. The claim of pericyte migration being due to delayed vascular development and not thyroid hormone seems too strong for me. Thyroid hormone metabolism is not addressed in these cells, and as they have been demonstrated to express MCT8, they could very well be expressing Dio3, to avoid T3 overloads in a particularly sensitive tissue, such as what happens in neurogenic niches in mammals.

This matter should be rephrased here or at least properly discussed in the discussion.

We have addressed this issue in the revised manuscript.

11. Normalize figure nomenclature with or without caps.

We have addressed this issue in the revised manuscript.

12. We 'wondered' if these cells?

We have addressed this issue in the revised manuscript.

13. I find Supp. Figure 5 of interest to be considered as main figure.

We agree with the reviewer; however, given the limited number of figures allowed in the main text, we had to make a choice. Instead of including the figure, we decided to present the graph that consolidates all this data. Additionally, including the figure in the main text would render the detailed images very small, which, in our view, would diminish the overall conclusions that readers could draw from them. Ultimately, we opted for this compromise, which we believe offers the best readability of our data.

14. Figure 5. Red instead of yellow arrowhead?

We have addressed this issue in the revised manuscript.

Discussion:

1. I would prefer "TH-signaling is essential for human vascularization". (As MCT8 transports both T3 and T4 and the study does not discard T4 actions).

We understand the reviewers' concerns. But again, I recall our previous answer to point 4 of the abstract. Moreover, in zebrafish embryos and brain organoids of AHDS patients, T3 impaired transport rather than T4 seems to be the pathophysiological origin of this condition, at least at the neurological level, which is what we are focusing on. This does not mean that in other cellular/tissue contexts, T4 transport is not impaired. However, this was not the aim of the present work.

2. "also support the persistent consequences of mct8 knockdown further confirming the observations made with the MCT8 knockdown model" seems a bit redundant, please rephrase.

We have addressed this issue in the revised manuscript.

REVIEWERS' COMMENTS:

Reviewer #1 (Remarks to the Author):

I have no comments now.

Reviewer #3 (Remarks to the Author):

Dear reviewer, first and foremost, thank you for your deep analysis of the data and the recognition of the importance for the field. I truly enjoyed this discussion, as, given the size of the field, it is not always possible. I completely understand your points and the reasoning behind them.

First, I want to thank the authors for thoroughly revising the multiple points from my previous review of the manuscript.

For me there is only one issue left that I would like to address, not only as a suggestion, but also as a discussion topic of value for the MCT8 community.

Aside from this, I find the manuscript form has been improved, and that this well-rounded study will be of interest to the field.

Abstract:

4. In humans MCT8 also transports T4.

In zebrafish it has been clearly shown after in vitro assays that at zebrafish physiological temperature (28.5C) zebrafish MCT8 can only transport T3 (PMID: 21952246). The biogenesis of thyroid hormones only starts at ~60hpf (PMID: 23022354). Furthermore, zebrafish eggs have available thyroid hormones before fertilization (PMID: 22379985). Collectively, this data proves that thyroid hormones, and namely T3, available for the embryo up until 48hpf (the oldest timepoint analyzed in our study) only has maternal origin. Although in vitro assays with human MCT8 have been shown to be able to transport both T4 and T3, a recent study using brain organoids derived from AHDS patients demonstrates that mutated human MCT8 specifically impairs T3 transport in those cells, further arguing that impaired T3 transport is the physiological problem that gives rise to AHDS (PMID: 38376950). We further strengthen these issues in the revised manuscript

First of all, thank you for the changes in the abstract regarding this issue, it fits better to me. I just wanted to discuss this claim a bit. They do not rule out T4 alterations. They prove T3, that is sure. However, when using DIO2 as a proxy to T4, they acknowledge the fact that they have a problem with CRYM sequestering T4 into the cells, and although

they prove that point, they do not pursue the experiment with/without MCT8, so they show no real information about T4 transport or action. This impedes ruling out T4 as a factor.

My problem with this argument is that it seems to consider that all these molecular events occur within the same cells. What has been extensively shown by various studies is that the production of T4 to T3 in the brain primarily takes place in glial cells, most likely astrocytes, which appear to use OATP1C1 as a gateway for the entry of T4, while MCT8 carries out the exit of produced T3. Supporting this physiological aspect of TH availability in the brain is the fact that D2 is expressed in glial cells, and D3 is expressed in neurons. This argues that the production of T3 and its export to target cells is critically controlled. In reference 13, it is clearly stated that CRYM serves as a reservoir for T3 originating from T4 to T3 conversion, and that an increase in CRYM parallels the increase in D2 activity. Moreover, this T3-i125 could only be detected in sonicated extracts and not in the medium alone. Therefore, my understanding of this data is that since D2 T4-T3 conversion does not occur in the same compartment or cell where T3 acts, it leads to an accumulation of T3 in CRYM within the cytosol of D2-expressing cells. It does not provide any information regarding these cells' ability to export T3 into target cells, which is likely impaired due to MCT8 mutations. I believe that such a scenario is plausible in human patients, where the accumulation of T3 from T4 in producing cells leads to elevated intracellular T3 levels due to impaired export, and therefore stimulates an increase in CRYM to buffer free T3 in producing cells that cannot export it efficiently to target cells. In fact, the decrease in D3 activity in organoids seems to further support this interpretation, as D3 is exclusively expressed in T3-target cells, suggesting no T3 is entering them, thus leading to decreased D3 activity. I think we cannot rule out T4; however, what seems increasingly compelling in this scenario is that MCT8 primarily transports T3 in the brain. Unfortunately, the authors of reference 13 did not measure OATP1C1 expression.

However, given that in zebrafish D2 or OATP1C1 are not expressed in the CNS during the analyzed stages, this further supports our claim that T3 is the only TH metabolite involved in the developmental action observed in our models. Moreover, it is compelling that the differentially expressed genes between WT and MCT8-deficient CO are similar to those we have previously reported, indicating a similar biological action of T3 and comparable developmental outcomes in zebrafish and humans. In my view, this further highlights the usefulness of zebrafish for studying T3 action in neurodevelopment, given that it represents a more simplified version of the physiology than that observed in humans. Additionally, as observed in zebrafish and in MCT8-deficient organoids, impaired T3 transport "does not impair neurodevelopment but only subsequent proliferation and differentiation" as we have previously reported in zebrafish.

Introduction:

6. Again, as in the abstract, MT3 transport is not the only molecular event behind AHDS, please rephrase.

We have addressed this issue in the revised manuscript. Please also refer to our comment on Abstract point 4.

So continuing with my previous reply, even though I find the phrase better this way, I still find the T3-only explanation rather incomplete, and even the authors of the cited reference seem to think so, as their statement is much milder, giving it as a potential key factor:

“In conclusion, here we provided definitive evidence that MCT8 is critical for early neurogenesis, mediating the bulk of the T3 uptake into developing human neural cells. Such perturbations and altered mechanisms likely play a key role in the pathogenesis of AHDS.”

I understand the reviewer’s point, but I interpret the conclusion of reference 13 differently. They seem to imply that MCT8 primarily takes up T3, but they cannot rule out that T3 can originate in specific brain cells via T4 from previous D2 conversion. In fact, their observations indicate that in neural progenitor cells, which express both D2 and MCT8 (and likely OATP1C1), there is accumulation of T3 bound to CRYM as a means to increase intracellular T3, allowing neural progenitor cells to survive and differentiate. Irrespective of this, given that the authors did not test the latter hypothesis, what seems clear from their study is that even if T3 is produced in brain cells, it does not appear to be able to reach target cells and accumulates, likely associated with CRYM. In this regard, I completely agree with the reviewer. As I stated previously, it seems very plausible that in the human brain of MCT8 patients, there is T3, arising from T4, but it cannot reach the target cells; it remains in the D2 T3-producing cells.

Discussion:

1. I would prefer "TH-signaling is essential for human vascularization". (As MCT8 transports both T3 and T4 and the study does not discard T4 actions).

We understand the reviewers’ concerns. But again, I recall our previous answer to point 4 of the abstract. Moreover, in zebrafish embryos and brain organoids of AHDS patients, T3 impaired transport rather than T4 seems to be the pathophysiological origin of this condition, at least at the neurological level, which is what we are focusing on. This does not mean that in other cellular/tissue contexts, T4 transport is not impaired. However, this was not the aim of the present work.

Referring to my previous reply, Salas-Lucia and colleagues do not rule out a T4 factor. However, I would like to add some more data to back my disagreement. In reference 9, you can find fetal brain TH levels in an MCT8 deficient patient postmortem sample (which also adds some human in vivo context): “Tissue-specific alteration of iodothyronines and their deiodinases. T4, T3, and rT3 concentrations were reduced by 50% in the cerebral cortex of the MCT8-deficient fetus, but T3 and T4 were normal in the liver. The hormonal deficiency in brain produced the expected increase in type 2 deiodinase and decrease in type 3 deiodinase mRNA expression (Supplemental Table 1).”

Thus, in this same cellular-tissular context, you have altered T4 and DIO2. I do not deny that T3 is a key factor, even the most important, but T4 cannot be ruled out as a potential factor contributing to AHDS.

Our zebrafish studies indicate an ancestral function for TH in neurodevelopment, suggesting that MT3 is likely the metabolite involved, with T4 playing a minimal role if any. This is supported by the absence of D2 expression and the lack of OATP1C1 up to hatching at around 48 hpf. I believe this also holds true for humans; however, here the evidence indicates that part of the T3 involved in neurodevelopment may originate from T4, which is converted in cells that express D2/OATP1C1, subsequently supplying T3 to target cells through MCT8. This interpretation suggests that AHDS arises not from T4 import or conversion to T3 but, as noted in reference 13, from inadequate T3 uptake by target cells, which disrupts GRN networks that depend on T3 for essential neurodevelopmental events. I think the observations in reference 9, mentioned by the reviewer, illustrate the same point emphasized in reference 13. In my opinion, this indicates that T3 transport is the necessary and sufficient factor for the development of AHDS. Nonetheless, I agree that the potential involvement of T4 transport in AHDS development cannot be dismissed, especially since this has not been thoroughly tested. However, recent findings from OATP1C1 deficient mice-only suggest differences compared to MCT8/OATP1C1 double KO mice, indicating that hormone entry pathway conditions its action, as demonstrated by in vitro experiments by Morte et al.

To accommodate the reviewers concerns we have included a line in the discussion regarding the possibility that at least in humans T4 transport could be affected as well in AHDS.